# HoloNets: Spectral Convolutions do extend to Directed Graphs

**Christian Koke & Daniel Cremers**
Technical University Munich and Munich Center for Machine Learning
{christian.koke,cremers}@tum.de

## Abstract

Within the graph learning community, conventional wisdom dictates that spectral convolutional networks may only be deployed on undirected graphs: Only there could the existence of a well-defined graph Fourier transform be guaranteed, so that information may be translated between spatial- and spectral domains. Here we show this traditional reliance on the graph Fourier transform to be superfluous and – making use of certain advanced tools from complex analysis and spectral theory – extend spectral convolutions to directed graphs. We provide a frequency-response interpretation of newly developed filters, investigate the influence of the basis used to express filters and discuss the interplay with characteristic operators on which networks are based. In order to thoroughly test the developed theory, we conduct experiments in real world settings, showcasing that directed spectral convolutional networks provide new state of the art results for heterophilic node classification on many datasets and – as opposed to baselines – may be rendered stable to resolution-scale varying topological perturbations. Our code is available at https://github.com/ChristianKoke/HoloNets.

## 1 Introduction

A particularly prominent line of research for graph neural networks is that of spectral convolutional architectures. These are among the theoretically best understood graph learning methods (Levie et al., 2019a; Ruiz et al., 2021a; Koke, 2023) and continue to set the state of the art on a diverse selection of tasks (Bianchi et al., 2019; He et al., 2021; 2022a; Wang & Zhang, 2022b). Furthermore, spectral interpretations allow to better analyse expressivity (Balcilar et al., 2021), shed light on shortcomings of established models (NT & Maehara, 2019) and guide the design of novel methods (Bo et al., 2023).

Traditionally, spectral convolutional filters are defined making use of the notion of a graph Fourier transform: Fixing a self-adjoint operator on an undirected $N$-node graph – traditionally a suitably normalized graph Laplacian $L = U^\top \Lambda U$ with eigenvalues $\Lambda = \mathrm{diag}(\lambda_1, ..., \lambda_N)$ – a notion of Fourier transform is defined by projecting a given signal $x$ onto the eigenvectors of $L$ via $x \mapsto Ux$. Since $L$ is self-adjoint, the eigenvectors form a complete basis and no information is lost in the process.

In analogy with the Euclidean convolution theorem, early spectral networks then defined convolution as multiplication in the "graph-Fourier domain" via $x \mapsto U^\top \cdot \mathrm{diag}(\theta_1, ..., \theta_N) \cdot Ux$, with learnable parameters $\{\theta_1, ..., \theta_N\}$ (Bruna et al., 2014). To avoid calculating an expensive explicit eigendecomposition $U$, Defferrard et al. (2016) proposed to instead parametrize graph convolutions via $x \mapsto U^\top g_\theta(\Lambda)Ux$, with $g_\theta$ a learnable scalar function applied to the eigenvalues $\Lambda$ as $g_\theta(\Lambda) = \mathrm{diag}\left(g_\theta(\lambda_1), ..., g_\theta(\lambda_N)\right)$. This precisely reproduces the mathematical definition of applying a scalar function $g_\theta$ to a self-adjoint operator $L$, so that choosing $g_\theta$ to be a (learnable) polynomial allowed to implement filters computationally much more economically as $g_\theta(L) = \sum_{k=1}^{K} \theta_k L^k$. Follow up works then considered the influence of the basis in which filters $\{g_\theta\}$ are learned (He et al., 2021; Levie et al., 2019b; Wang & Zhang, 2022a) and established that such filters provide networks with the ability to generalize to unseen graphs (Levie et al., 2019a; Ruiz et al., 2021b; Koke, 2023).

Common among all these works, is the need for the underlying graph to be undirected: Only then are the corresponding operators symmetric, so that a complete set of orthogonal eigenvectors exists and the graph Fourier transform $U$ (used to define the filter $g_\theta(L)$ via $x \mapsto Ug_\theta(\Lambda)U^\top x$) is well-defined.[1]

---

[1] Strictly speaking it is not symmetry ($L = L^\top$) but normality ($LL^\top = L^\top L$) of $L$ that ensures this.

Currently however, the graph learning community is endeavouring to finally also account for the previously neglected directionality of edges, when designing new methods (Zhang et al., 2021; Rossi et al., 2023; Beaini et al., 2021; Geisler et al., 2023; He et al., 2022b). Since characteristic operators on digraphs are generically not self-adjoint, traditional spectral approaches so far remained inaccessible in this undertaking. Instead, works such as Zhang et al. (2021); He et al. (2022b) resorted to limiting themselves to certain specialized operators able to preserve self-adjointness in this directed setting. While this approach is not without merit, the traditional adherence to the graph Fourier transform remains a severely limiting factor when attempting to extend spectral networks to directed graphs.

**Contributions:** In this paper we argue to completely dispose with this reliance on the graph Fourier transform and instead take the concept of learnable functions applied to characteristic operators as fundamental. This conceptual shift allows us to consistently define spectral convolutional filters on directed graphs. We provide a corresponding frequency perspective, analyze the interplay with chosen characteristic operators and discuss the importance of the basis used to express these novel filters. The developed theory is thoroughly tested on real world data: It is found that directed spectral convolutional networks provide new state of the art results for heterophilic node classification and – as opposed to baselines – may be rendered stable to resolution-scale varying topological perturbations.

## 2 SIGNAL PROCESSING ON DIRECTED GRAPHS

Our work is mathematically rooted in the field of graph signal processing; briefly reviewed here:

**Weighted directed graphs:** A directed graph $G := (\mathcal{G}, \mathcal{E})$ is a collection of nodes $\mathcal{G}$ and edges $\mathcal{E} \subseteq \mathcal{G} \times \mathcal{G}$ for which $(i, j) \in \mathcal{E}$ does not necessarily imply $(j, i) \in \mathcal{E}$. We allow nodes $i \in \mathcal{G}$ to have individual node-weights $\mu_i > 0$ and generically assume edge-weights $w_{ij} \geqslant 0$ not necessarily equal to unity or zero. In a social network, a **node weight** $\mu_i = 1$ might signify that a node represents a single user, while a weight $\mu_j > 1$ would indicate that node $j$ represents a group of users. Similarly, **edge weight**s $\{w_{ij}\}$ could be used to encode how many messages have been exchanged between nodes $i$ and $j$. Importantly, since we consider directed graphs, we generically have $w_{ij} \neq w_{ji}$.

Edge weights also determine the so called **reaches** of a graph, which generalize the concept of connected components of undirected graphs (Veerman & Lyons, 2020): A subgraph $R \subseteq G$ is called reach, if for any two vertices $a, b \in R$ there is a directed path in $R$ along which the (directed) edge weights do not vanish, and $R$ simultaneously possesses no outgoing connections (i.e. for any $c \in G$ with $c \notin R$: $w_{ca} = 0$). For us, this concept will be important in generalizing the notion of scale insensitive networks (Koke et al., 2023) to directed graphs in Section 3.3 below.

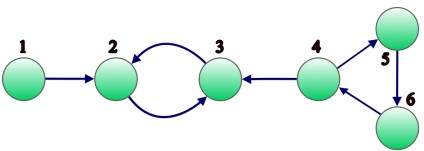

Figure 1: A di-graph with reaches $R_1 = \{1, 2, 3\}$ and $R_2 = \{3, 4, 5, 6, 2\}$.

**Feature spaces:** Given $F$-dimensional node features on a graph with $N = |\mathcal{G}|$ nodes, we may collect individual node-feature vectors into a feature matrix $X$ of dimension $N \times F$. Taking into account our node weights, we equip the space of such signals with an inner-product according to $\langle X, Y \rangle = \text{Tr}(X^* M Y) = \sum_{i=1}^{N} \sum_{j=1}^{F} (\overline{X}_{ij} Y_{ij}) \mu_i$ with $M = \text{diag}(\{\mu_i\})$ the diagonal matrix of node-weights. Here $X^*$ denotes the (hermitian) adjoint of $X$ (c.f. Appendix B for a brief recapitulation). Associated to this inner product is the standard 2-norm $\|X\|_2^2 = \sum_{i=1}^{N} \sum_{j=1}^{F} |X_{ij}|^2 \mu_i$.

**Characteristic Operators:** Information about the geometry of a graph is encapsulated into the set of edge weights, collected into the weight matrix $W$. From this, the diagonal in-degree and out-degree matrices ($D_{ii}^{\text{in}} = \sum_j W_{ij}$, $D_{jj}^{\text{out}} = \sum_i W_{ij}$) may be derived. Together with the node-weight matrix $M$ defined above, various characteristic operators capturing the underlying geometry of the graph may then be constructed. Relevant to us – apart from the weight matrix $W$ – will especially be the (in-degree) Laplacian $L^{\text{in}} := M^{-1}(D^{\text{in}} - W)$, which is intimately related to consensus and diffusion on directed graphs (Veerman & Kummel, 2019). Importantly, such characteristic operators $T$ are generically not self-adjoint. Hence they do not admit a complete set of orthogonal eigenvectors and their spectrum $\sigma(T)$ contains complex eigenvalues $\lambda \in \mathbb{C}$. Appendix B contains additional details on such operators, their canonical (Jordan) decomposition and associated *generalized* eigenvectors.

## 3 SPECTRAL CONVOLUTIONS ON DIRECTED GRAPHS

Since characteristic operators on directed graphs generically do not admit a complete set of orthogonal eigenvectors, we cannot make use of the notion of a graph Fourier transform to consistently define filters of the form $g_\theta(T)$. While this might initially seem to constitute an insurmountable obstacle, the task of defining operators of the form $g(T)$ for a given operator $T$ and appropriate classes of scalar-valued functions $\{g\}$ – such that relations between the functions $\{g\}$ translate into according relation of the operators $\{g(T)\}$ – is in fact a well studied problem (Haase, 2006; Colombo et al., 2011). Corresponding techniques typically bear the name "functional calculus" and importantly are also definable if the underlying operator $T$ is not self-adjoint (Cowling et al., 1996):

### 3.1 THE HOLOMORPHIC FUNCTIONAL CALCULUS

In the undirected setting, it was possible to essentially apply arbitrary functions $\{g\}$ to the characteristic operator $T = U^\top \Lambda U$ by making use of the complete eigendecomposition as $g(T) := U^\top g_\theta(\Lambda)U$. However, a different approach to consistently defining the matrix $g(T)$ – not contingent on such a decomposition – is available if (and only if) one restricts $g$ to be a **holomorphic** function: For a given subset $U$ of the complex plane, these are the complex valued functions $g : U \to \mathbb{C}$ for which the complex derivative $dg(z)/dz$ exists everywhere on the domain $U$ (c.f. Appendix D for more details).

The property of holomorphic functions that allows to consistently define the matrix $g(T)$ is the fact that any function value $g(\lambda)$ can be reproduced by calculating an integral of the function $g$ along a path $\Gamma$ encircling $\lambda$ (c.f. also Fig. 2) as

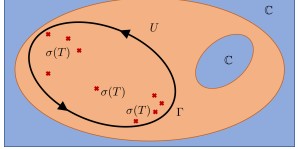

$$g(\lambda) = -\frac{1}{2\pi i} \oint_\Gamma g(z) \cdot (\lambda - z)^{-1} dz. \qquad (1)$$

Figure 2: Cauchy Integral (1)

Here "$dz$" denotes the complex line-integration-measure in $\mathbb{C}$. In order to define the matrix $g(T)$, the formal replacement $\lambda \mapsto T$ is then made on both sides of (1), with the path $\Gamma$ now not only encircling a single value $\lambda$ but all eigenvalues $\lambda \in \sigma(T)$ (c.f. also Fig. 3):

$$g(T) := -\frac{1}{2\pi i} \oint_\Gamma g(z) \cdot (T - z \cdot Id)^{-1} dz \qquad (2)$$

Note that $(T - z \cdot Id)^{-1}$ – and hence the integral in (2) – is indeed well-defined: All eigenvalues of $T$ are assumed to lie *inside* the path $\Gamma$. For any choice of integration variable $z$ *on* this path $\Gamma$, the matrix $(T - z \cdot Id)$ is thus indeed invertible, since $z$ is never an eigenvalue.

Figure 3: Operator Integral (2)

The integral in (2) defines what is called the **holomorphic functional calculus** (Gindler, 1966; Kato, 1976). Importantly (c.f. Appendix E), the definition of $g(T)$ in (2) agrees with algebraic relations:

**Theorem 3.1.** Applying a polynomial $g(\lambda) := \sum_{k=0}^K a_k \lambda^k$ to $T$ yields $g(T) = \sum_{k=0}^K a_k T^k$. Similarly applying the function $g(\lambda) = 1/\lambda$ yields $g(T) = T^{-1}$ provided $T$ is invertible.

### 3.2 SPECTRAL CONVOLUTIONAL FILTERS ON DIRECTED GRAPHS

Since the holomorphic functional calculus is evidently no longer contingent on $T$ being self-adjoint, it indeed provides an avenue to consistently define spectral convolutional filters on directed graphs.

**Parametrized spectral convolutional filters:** In practice it is of course prohibitively expensive to continuously compute the integral (2) as the learnable function $g$ is updated during training. Instead, we propose to represent a generic holomorphic function $g$ via a set of basis functions $\{\Psi_i\}_{i \in I}$ as $g_\theta(z) := \sum_{i \in I} \theta_i \cdot \Psi_i(z)$ with learnable coefficients $\{\theta_i\}_{i \in I}$ parametrizing the filter $g_\theta$.

For the 'simpler' basis functions $\{\Psi_i\}_{i \in I}$, we either precompute the integral (2), or perform it analytically (c.f. Section 3.3 below). During training and inference the matrices $\Psi_i(T) \equiv -\frac{1}{2\pi i} \oint_\Gamma \Psi_i(z) \cdot (T - z \cdot Id)^{-1} dz$ are then already computed and learnable filters are given as

$$g_\theta(T) := \sum_{i \in I} \theta_i \cdot \Psi_i(T). \qquad (3)$$

Generically, each coefficient $\theta_i$ may be chosen as a *complex* number; equivalent to two *real* parameters. If the functions $\{\Psi_i\}_{i \in I}$ are chosen such that each matrix $\Psi_i(T)$ contains only real entries (e.g. for $\Psi$ a polynomial with real coefficients), it is possible to restrict convolutional filters to being purely real: In this setting, choosing the parameters $\{\theta_i\}$ to be purely real as well, leads to $g_\theta(T) = \sum_{i \in I} \theta_i \cdot \Psi_i(T)$ itself being a matrix that contains only real entries. In this way, complex numbers need never to appear within our network, if this is not desired. In Theorem 4.1 of Section 4 below, we discuss how, under mild and reasonable assumptions, such a complexity-reduction to using only real parameters can be performed without decreasing the expressive power of corresponding networks.

Irrespective of whether real or complex weights are employed, the utilized filter bank $\{\Psi_i\}_{i \in I}$ determines the space of learnable functions $g_\theta \in \mathrm{span}(\{\Psi_i\}_{i \in I})$ and thus contributes significantly to the inductive bias present in the network. It should thus be adjusted to the respective task at hand.

**The Action of Filters in the Spectral Domain:** In order to determine which basis functions are adapted to which tasks, a "frequency-response" interpretation of spectral filters is expedient:

In the **undirected setting** this proceeded by decomposing any characteristic operator $T$ into a sum $T = \sum_{\lambda \in \sigma(T)} \lambda \cdot P_\lambda$ over its distinct eigenvalues. The spectral action of any function $g$ was then given by $g(T) = \sum_{\lambda \in \sigma(T)} g(\lambda) \cdot P_\lambda$. Here the spectral projections $P_\lambda$ project each vector to the space spanned by the eigenvectors $\{v_i\}$ corresponding to the eigenvalue $\lambda$ (i.e. satisfying $(T - \lambda Id)v_i = 0$).

In the **directed setting**, there only exists a basis of *generalized* eigenvectors $\{w_i\}_{i=1}^N$; each satisfying $(T - \lambda \cdot Id)^m w_i = 0$ for some $\lambda \in \sigma(T)$ and $m \in \mathbb{N}$ (c.f. Appendix B). Denoting by $P_\lambda$ the matrix projecting onto the space spanned by these *generalized* eigenvectors associated to the eigenvalue $\lambda \in \sigma(T)$, any operator $T$ may be written as[2] $T = \sum_{\lambda \in \sigma(T)} \lambda \cdot P_\lambda + \sum_{\lambda \in \sigma(T)} (T - \lambda \cdot Id) \cdot P_\lambda$. It can then be shown (Kato, 1976), that the spectral action of a given function $g$ is given as

$$g(T) = \sum_{\lambda \in \sigma(T)} g(\lambda) P_\lambda + \sum_{\lambda \in \sigma(T)} \left[ \sum_{n=1}^{m_\lambda - 1} \frac{g^{(n)}(\lambda)}{n!} (T - \lambda \cdot Id)^n \right] P_\lambda. \tag{4}$$

Here the number $m_\lambda$ is the algebraic multiplicity of the eigenvalue $\lambda$; i.e. the dimension of the associated *generalized* eigenspace. The notation $g^{(n)}$ denotes the $n^{\text{th}}$ complex derivative of $g$. The appearance of such derivative terms in (4) is again evidence, that we indeed needed to restrict from generic- to differentiable functions[3] in order to sensibly define directed spectral convolutional filters.

It is instructive to gain some intuition about the second sum on the right-hand-side of the frequency response (4), as it is not familiar from undirected graphs (since it vanishes if $T$ is self-adjoint): As an example consider the un-weighted directed path graph on three nodes depicted in Fig. 4 and choose as characteristic operator $T$ the adjacency matrix (i.e. $T = W$). It is not hard to see (c.f. Appendix C for an explicit calculation) that the only eigenvalue of $W$ is given by $\lambda = 0$ with algebraic multiplicity $m_\lambda = 3$. Since spectral projections always satisfy $\sum_{\lambda \in \sigma(T)} P_\lambda = Id$ (c.f. Appendix B), and here $\sigma(W) = \{0\}$ we thus have $P_{\lambda=0} = Id$ in this case.

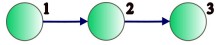

Figure 4: Directed path on 3 nodes

Suppose now we are tasked with finding a (non-trivial) holomorphic filter $g(\lambda)$ such that $g(T) = 0$. The right-hand sum in (4) implies, that beyond $g(0) = 0$, also the first and second derivative of $g(\lambda)$ needs to vanish at $\lambda = 0$ to achieve this. Hence the zero of $g(\lambda)$ at $\lambda = 0$ must be at least of order three; or equivalently for $\lambda \to 0$ we need $g(\lambda) = o(\lambda^3)$. This behaviour is of course exactly mirrored in the spatial domain: As applying $W$ simply moves information at a given node along the path, applying $W$ once or twice still leaves information present. After two applications, only node 3 still contains information and thus applying $W^k$ precisely removes all information if and only if $k \geqslant 3$. Without the assumption of acyclicity, the spectrum of characteristic operators of course generically does not consist only of the eigenvalue $\lambda = 0$. Thus generically $P_{\lambda=0} \neq Id$ and the role played by the operator $T = W$ in the considerations above is instead played by its restriction $(T \cdot P_{\lambda=0})$ to the generalized eigenspace corresponding to the eigenvalue $\lambda = 0$.

For us, the spectral response (4) provides guidance when considering scale-insensitive convolutional filters on *directed* graphs in Sections 3.3 and 4 below. The spectral response (4) is however never used to *implement* filters: As discussed above, this is achieved much more economically via (3).

---

[2]Additional details on this so called Jordan Chevalley decomposition are provided in Appendix B.

[3]N.B.: A once-complex-differentiable function is automatically infinitely often differentiable (Ahlfors, 1966).

### 3.3 Explicit Filter Banks

Having laid the theoretical foundations, we consider examples of task-adapted filter banks $\{\Psi_i\}_{i \in I}$.

#### 3.3.1 Bounded Spectral Domain: Faber Polynomials

First, let us consider spectral networks on a single graph with a fixed characteristic operator $T$. From the holomorphic functional calculus (2), we infer that convolutional filters $\{g(T)\}$ are in principle provided by all holomorphic functions $\{g\}$ defined on a domain $U$ which contains all eigenvalues $\lambda \in \sigma(T)$ of $T$. As noted above, implementing an arbitrary holomorphic $g$ is however too costly, and we instead approximate $g$ via a collection of simpler basis functions $\{\Psi_i\}_{i \in I}$ as $g(\lambda) \approx \sum_{i \in I} \theta_i \Psi_i(\lambda)$.

In order to choose the filter bank $\{\Psi_i\}_{i \in I}$, we thus need to answer the question of how to optimally approximate arbitrary holomorphic functions on a given fixed domain $U$. The solution to this problem is given in the guise of **Faber polynomials** (Ellacott, 1983; Coleman & Smith, 1987) which generalize the familiar Chebychev polynomials utilized in (Defferrard et al., 2016) to subsets $U$ of the complex plane (Elliott, 1978). Faber polynomials provide near mini-max polynomial approximation[4] to any holomorphic function defined on a domain $U$ satisfying some minimal conditions (c.f. Elliott (1978) for exact details). What is more, they have already successfully been employed in numerically approximating matrices of the form $g(T)$ for $T$ not necessarily symmetric (Moret & Novati, 2001).

While for a generic domain $U$ Faber polynomials are impossible to compute analytically, this poses no limitations to us in practice: Short of a costly explicit calculation of the spectrum $\sigma(T)$, the only information that is generically available, is that eigenvalues may be located anywhere within a circle of radius $\|T\|$. This circle must thus be contained in any valid domain $U$. Making the minimal choice by taking $U$ to be exactly this circle, the $n^{\text{th}}$-Faber polynomial may be analytically calculated (He, 1995): Up to normalization (absorbed into the learnable parameters) it is given by the monomial $\lambda^n$. We thus take our $n^{\text{th}}$ basis element $\Psi_n(\lambda)$ to be given precisely by this monomial: $\Psi_n(\lambda) = \lambda^n$. Thus Faber polynomials evaluate to $\Psi_k(T) = T^k$ on our characteristic operator $T$ (c.f. Thoerem 3.1). In a setting where more detailed information on $\sigma(T)$ is available, the domain $U$ may of course be adapted to reflect this. Corresponding Faber polynomials might then be pre-computed numerically.

#### 3.3.2 Unbounded Spectral Domain: Functions decaying at complex infinity

In the multi-graph setting – e.g. during graph classification – we are confronted with the possibility that distinct graphs may describe the same underlying object (Levie et al., 2019a; Maskey et al., 2021; Koke, 2023). This might for example occur if two distinct graphs discretize the same underlying continuous space; e.g. at different resolution scales. In this setting – instead of precise placements of nodes – what is actually important is the overall structure and geometry of the respective graphs.

Un-normalized Laplacians provide a convenient multi-scale descriptions of such graphs, as they encode information corresponding to coarse geometry into small (in modulus) eigenvalues, while finer graph structures correspond to larger eigenvalues (Chung, 1997; Ng et al., 2001). When designing networks whose outputs are not overly sensitive to fine print articulations of graphs, the spectral response (4) then provides guidance on deterimng which holomorphic filters $g$ are able to suppress this superfluous high-lying spectral information: It is sufficient that $g^{(n)}(\lambda)/n! \approx 0$ for $|\lambda| \gg 1$. It can be shown that no holomorphic function with such large-$|\lambda|$ asymptotics defined on all of $\mathbb{C}$ exists.[5] We thus make the minimal necessary change and assume $g$ to be defined on a *punctured* domain $U = \mathbb{C}\backslash\{y\}$ instead. The choice of $y \in \mathbb{C}$ is treated as a hyperparameter, which may be adjusted to the task at hand. Any such $g$ may then be expanded as $g(\lambda) = \sum_{j=1}^{\infty} \theta_j (\lambda - y)^{-j}$ for some coefficients $\{\theta_i\}_{i=1}^{\infty}$ (Bak & Newman, 2017). Evaluating the defining integral (2) for the Laplacian $L^{\text{in}}$ on the atoms $\Psi_j(\lambda) = (\lambda - y)^{-j}$ yields $\Psi_j(L^{\text{in}}) = ([L^{\text{in}} - y \cdot Id]^{-1})^j$; as proved in Appendix E. Hence corresponding filters are polynomials in the **resolvent** $R_y(L^{\text{in}}) := [L^{\text{in}} - y \cdot Id]^{-1}$ of $L^{\text{in}}$.

Such resolvents are traditionally used as tools to compare operators with potentially divergent norms (Teschl, 2014). Recently Koke et al. (2023) utilized them in the *undirected* setting to construct networks provably assigning similar feature-vectors to weighted graphs describing the same underlying object at different resolution-scales. Our approach extends these networks to the *directed* setting:

---

[4] I.e. minimizing the maximal approximation error on the domain of definition $U$.

[5] This is an immediate consequence of Liouville's theorem in complex analysis (Ahlfors, 1966).

**Effective *directed* Limit Graphs:**  From a diffusion perspective, information in a graph equalizes faster along edges with large weights. In the limit where the edge-weights within certain sub-graphs tend to infinity, information within these clusters equalizes immediately and such sub-graphs should thus effectively behave as single nodes. Extending *undirected*-graph results (Koke & Kutyniok, 2022; Koke et al., 2023), we here establish rigorously that this is indeed also true in the *directed* setting.

Mathematically, we make our arguments precise by considering a graph $G$ with a weight matrix $W$ admitting a (disjoint) two-scale decomposition as $W = W^{\text{regular}} + c \cdot W^{\text{high}}$ (c.f. Fig. 5). As the larger weight scale $c \gg 1$ tends to infinity, we then establish that the resolvent $R_y(L^{\text{in}})$ on $G$ converges to the resolvent $R_y(\underline{L}^{\text{in}})$ of the Laplacian $\underline{L}^{\text{in}}$ on a coarse-grained limit graph $\underline{G}$. This limit $\underline{G}$ arises by collapsing the reaches $R$ of the graph $G_{\text{high}} = (G, W^{\text{high}})$ (c.f. Fig. 5 (c)) into single nodes. For technical reasons, we here assume[6] equal in- and out-degrees within $G_{\text{high}}$ (i.e. $\sum_i W_{ij}^{\text{high}} = \sum_i W_{ji}^{\text{high}}$). Appendix G contains proofs corresponding to the results below.

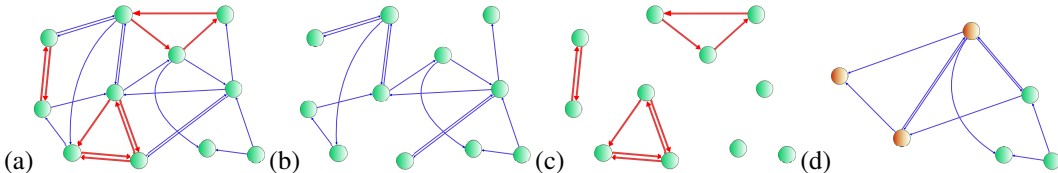

Figure 5: (a) Graph $G$ with $W^{\text{regular}}$ (blue) & $W^{\text{high}}$ (red); (b) $W^{\text{regular}}$; (c) $W^{\text{high}}$; (d) Limit Graph $\underline{G}$

When defining $\underline{G}$, *directed reaches* now replace the *undirected components* of Koke et al. (2023):
**Definition 3.2.** The node set $\underline{\mathcal{G}}$ of $\underline{G}$ is constituted by the set of all reaches in $G_{\text{high}}$. Edges $\underline{\mathcal{E}}$ of $\underline{G}$ are given by those elements $(R, P) \in \underline{\mathcal{G}} \times \underline{\mathcal{G}}$ with non-zero agglomerated edge weight $\underline{W}_{RP} = \sum_{r \in R} \sum_{p \in P} W_{rp}$. Node weights in $\underline{G}$ are defined similarly by aggregating as $\underline{\mu}_R = \sum_{r \in R} \mu_r$.

To map signals between these graphs, translation operators $J^{\downarrow}, J^{\uparrow}$ are needed. Let $x$ be a scalar graph signal and let $\mathbb{1}_R$ be the vector that has 1 as entry for nodes $r \in R$ and zero otherwise. Denote by $u_R$ the entry of $u$ at node $R \in \underline{\mathcal{G}}$. The projection operator $J^{\downarrow}$ is then defined component-wise by evaluation at node $R \in \underline{\mathcal{G}}$ as $(J^{\downarrow}x)_R = \langle \mathbb{1}_R, x \rangle / \underline{\mu}_R$. Interpolation is defined as $J^{\uparrow}u = \sum_{R \in \underline{\mathcal{G}}} u_R \cdot \mathbb{1}_R$. The maps $J^{\uparrow}, J^{\downarrow}$ are then extended from single features $\{x\}$ to feature *matrices* $\{X\}$ via linearity.

With these preparations, we can now rigorously establish the suspected effective behaviour of clusters:
**Theorem 3.3.** In the above setting, we have $\|R_y(L^{\text{in}}) \cdot X - J^{\uparrow}R_y(\underline{L}^{\text{in}})J^{\downarrow} \cdot X\| \longrightarrow 0$ as $c \to \infty$.

For $c \gg 1$, applying the resolvent $R_y(L^{\text{in}})$ on $G$ is thus essentially the same as first projecting to the coarse-grained graph $\underline{G}$ (where all strongly connected clusters are collapsed), applying the corresponding resolvent there and then interpolating back up to $G$. The geometric information within $R_y(L^{\text{in}})$ is thus essentially reduced to that of the coarse grained geometry within $\underline{G}$.

Large weights within a graph typically correspond to fine-structure articulations of its geometry: For graph-discretisations of continuous spaces, edge weights e.g. typically correspond to inverse discretization lengths ($w_{ij} \sim 1/d_{ij}$) and strongly connected clusters describe closely co-located nodes. In social-networks, edge weights might encode a closeness-measure, and coarse-graining would correspond to considering interactions between (tightly connected) communities as opposed to individual users. In either case, fine print articulations are discarded when applying resolvents.

**Stability of Filters:**  This reduction to a limit description on $\underline{G}$ is respected by our filters $\{g_\theta\}$:
**Theorem 3.4.** In the above setting, we have $\|g_\theta(L^{\text{in}}) \cdot X - J^{\uparrow}g_\theta(\underline{L}^{\text{in}})J^{\downarrow} \cdot X\| \longrightarrow 0$ as $c \to \infty$.

If the weight-scale $c$ is very large, applying the learned filter $g_\theta(\lambda) = \sum_{i=1}^{K} \theta_i(\lambda - y)^{-i}$ to a signal $X$ on $G$ as $X \mapsto g_\theta(L^{\text{in}}) \cdot X$ thus is essentially the same as first discarding fine-structure information by projecting $X$ to $\underline{G}$, applying the spectral filter $g_\theta$ there and subsequently interpolating back to $G$. Information about the precise articulation of a given graph $G$ is thus suppressed in this propagation scheme; it is purely determined by the graph structure of the coarse-grained description $\underline{G}$. Theorem 4.2 below establishes that this behaviour persists for entire (directed) spectral convolutional networks.

---

[6]This is known as Kirchhoff's assumption (Balti, 2018); reproducing the eponymous law of electrical circuits.

# 4 SPECTRAL NETWORKS ON DIRECTED GRAPHS: HOLONETS

We now collect holomorphic filters into corresponding networks; termed **HoloNets**. In doing so, we need to account for the possibility that given edge directionalities might limit the information-flow facilitated by filters $\{g_\theta(T)\}$: In the path-graph setting of Fig. 4 for example, a polynomial filter in the adjacency matrix would only transport information *along* the graph; features of earlier nodes would never be augmented with information about later nodes. To circumvent this, we allow for two sets of filters $\{g_\theta^{\text{fwd}}(T)\}$ and $\{g_\theta^{\text{bwd}}(T^*)\}$ based on the characteristic operator $T$ and its adjoint $T^*$. Allowing these forward- and backward-filters to be learned in different bases $\{\Psi_i^{\text{fwd/bwd}}\}_{i \in I^{\text{fwd/bwd}}}$, we may write $g_\theta^{\text{fwd/bwd}}(\lambda) = \sum_{i \in I^{\text{fwd/bwd}}} \theta_i^{\text{fwd/bwd}} \Psi_i^{\text{fwd/bwd}}(\lambda)$. With bias matrices $B^{\ell+1}$ of size $N \times F_{\ell+1}$ and weight matrices $W_k^{\text{fwd/bwd},\ell+1}$ of dimension $F_\ell \times F_{\ell+1}$, our update rule is then efficiently implemented as

$$X^\ell = \rho \left( \alpha \sum_{i \in I^{\text{fwd}}} \Psi_i^{\text{fwd}}(T) \cdot X^{\ell-1} \cdot W_i^{\text{fwd},\ell} + (1-\alpha) \sum_{i \in I^{\text{bwd}}} \Psi_i^{\text{bwd}}(T^*) \cdot X^{\ell-1} \cdot W_i^{\text{bwd},\ell} + B^\ell \right).$$

Here $\rho$ is a point-wise non-linearity, and the parameter $\alpha \in [0,1]$ – learnable or tunable – is introduced following Rossi et al. (2023) to allow for a prejudiced weighting of the forward or backward direction. We additionally provide a pseudocode description of the corresponding models in Appendix J.

The generically complex weights & biases may often be restricted to $\mathbb{R}$ without losing expressivity:

**Theorem 4.1.** Suppose for filter banks $\{\Psi_i^{\text{fwd/fwd}}\}_{I^{\text{fwd/fwd}}}$ that the matrices $\Psi_i^{\text{fwd}}(T)$, $\Psi_i^{\text{bwd}}(T^*)$ contain only real entries. Then any HoloNet with layer-widths $\{F_\ell\}$ with complex weights & biases and a non linearity that acts on complex numbers componentwise as $\rho(a+ib) = \widetilde{\rho}(a) + i\widetilde{\rho}(a)$ can be exactly represented by a HoloNet of widths $\{2 \cdot F_\ell\}$ utilizing $\widetilde{\rho}$ and employing only real weights & biases.

This result (proved in Appendix H) establishes that for the same number of *real* parameters, real HoloNets theoretically have greater expressive power than complex ones. In our experiments in Section 5 below, we empirically find that complex weights do provide advantages on some graphs. Thus we propose to treat the choice of complex vs. real parameters as a binary hyperparameter.

**FaberNet:**  The first specific instantiation of HoloNets we consider, employs the Faber Polynomials of Section 3.3.1 for both the forward and backward filter banks. Since Rossi et al. (2023) established that considering edge directionality is especially beneficial on heterophilic graphs, this is also our envisioned target for the corresponding networks. We thus use as characteristic operator a matrix that avoids direct comparison of feature vectors of a node with those of immediate neighbours: We choose $T = (D^{in})^{-\frac{1}{4}} \cdot W \cdot (D^{out})^{-\frac{1}{4}}$ since it has a zero-diagonal and its normalization performed well empirically. For the same reason of heterophily, we also consider the choice of whether to include the Faber polynomial $\Psi_0(\lambda) = 1$ in our basis as a hyperparameter. As non-linearity, we choose either $\rho(a+ib) = \text{ReLu}(a) + i\text{ReLu}(b)$ or $\rho(a+ib) = |a| + i|b|$. Appendix I contains additional details.

**Dir-ResolvNet:**  In order to build networks that are insensitive to the fine-print articulation of *directed* graphs, we take as filter bank the functions $\{\Psi_i(\lambda) = (\lambda - y)^{-i}\}_{i>0}$ evaluated on the Laplacian $L^{\text{in}}$ for both the forward and backward direction. To account for individual node-weights when building up graph-level features, we use an aggregation $\Omega$ that aggregates $N \times F$-dimensional node-feature matrices as $\Omega(X)_j = \sum_{i=1}^{N} |X_{ij}| \cdot \mu_i$ to a graph-feature $\Omega(X) \in \mathbb{R}^F$. Graph-level stability under varying resolution scales is then captured by our next result:

**Theorem 4.2.** Let $\Phi$ and $\underline{\Phi}$ be the feature maps associated to Dir-ResolvNets with the same weights and biases deployed on graphs $G$ and $\underline{G}$ as defined in Section 3.3.2. With $\Omega$ the aggregation method specified above and $W = W^{\text{regular}} + c \cdot W^{\text{high}}$ as in Theorem 3.4, we have for $c \to \infty$:

$$\| \Omega(\Phi(X)) - \Omega(\underline{\Phi}(J^\downarrow X)) \| \longrightarrow 0$$

Appendix G contains proofs of this and additional stability results. From Theorem 4.2 we conclude that graph-level features generated by a Dir-ResolvNet are indeed insensitive to fine print articulations of weighted digraphs: As discussed in Section 3.3.2, geometric information corresponding to such fine details is typically encoded into strongly connected sub-graphs; with the connection strength $c$ corresponding to the level of detail. However, information about the structure of these sub-graphs is precisely what is discarded when moving to $\underline{G}$ via $J^\downarrow$. Thus the greater the level of detail within $G$, the more similar are generated feature-vectors to those of a (relatively) coarse-grained description $\underline{G}$.

## 5 EXPERIMENTS

We present experiments on real-world data to evaluate the capabilities of our HoloNets numerically.

### 5.1 FABERNET: NODE CLASSIFICATION

We first evaluate on the task of node-classification in the presence of heterophily. We consider multiple heterophilic graph-datasets on which we compare the performance of our **FaberNet** instantiation of the HoloNet framework against a representative array of baselines: As *simple baselines* we consider MLP and GCN (Kipf & Welling, 2017). $H_2$GCN (Zhu et al., 2020), GPR-GNN (Chien et al., 2021), LINKX (Lim et al., 2021), FSGNN (Maurya et al., 2021), ACM-GCN (Luan et al., 2022), GloGNN (Li et al., 2022) and Gradient Gating (Rusch et al., 2023) constitute *heterophilic state-of-the-art models*. Finally *state-of-the-art models for directed graphs* are given by DiGCN (Tong et al., 2020), MagNet (Zhang et al., 2021) and Dir-GNN (Rossi et al., 2023). Appendix I contains dataset statistics as well as additional details on baselines, experimental setup and hyperparameters.

Table 1: Results on real-world directed heterophilic datasets. OOM indicates out of memory.

| Homophily | Squirrel 0.223 | Chameleon 0.235 | Arxiv-year 0.221 | Snap-patents 0.218 | Roman-Empire 0.05 |
|---|---|---|---|---|---|
| MLP | $28.77 \pm 1.56$ | $46.21 \pm 2.99$ | $36.70 \pm 0.21$ | $31.34 \pm 0.05$ | $64.94 \pm 0.62$ |
| GCN | $53.43 \pm 2.01$ | $64.82 \pm 2.24$ | $46.02 \pm 0.26$ | $51.02 \pm 0.06$ | $73.69 \pm 0.74$ |
| $H_2$GCN | $37.90 \pm 2.02$ | $59.39 \pm 1.98$ | $49.09 \pm 0.10$ | OOM | $60.11 \pm 0.52$ |
| GPR-GNN | $54.35 \pm 0.87$ | $62.85 \pm 2.90$ | $45.07 \pm 0.21$ | $40.19 \pm 0.03$ | $64.85 \pm 0.27$ |
| LINKX | $61.81 \pm 1.80$ | $68.42 \pm 1.38$ | $56.00 \pm 0.17$ | $61.95 \pm 0.12$ | $37.55 \pm 0.36$ |
| FSGNN | $74.10 \pm 1.89$ | $78.27 \pm 1.28$ | $50.47 \pm 0.21$ | $65.07 \pm 0.03$ | $79.92 \pm 0.56$ |
| ACM-GCN | $67.40 \pm 2.21$ | $74.76 \pm 2.20$ | $47.37 \pm 0.59$ | $55.14 \pm 0.16$ | $69.66 \pm 0.62$ |
| GloGNN | $57.88 \pm 1.76$ | $71.21 \pm 1.84$ | $54.79 \pm 0.25$ | $62.09 \pm 0.27$ | $59.63 \pm 0.69$ |
| Grad. Gating | $64.26 \pm 2.38$ | $71.40 \pm 2.38$ | $63.30 \pm 1.84$ | $69.50 \pm 0.39$ | $82.16 \pm 0.78$ |
| DiGCN | $37.74 \pm 1.54$ | $52.24 \pm 3.65$ | OOM | OOM | $52.71 \pm 0.32$ |
| MagNet | $39.01 \pm 1.93$ | $58.22 \pm 2.87$ | $60.29 \pm 0.27$ | OOM | $88.07 \pm 0.27$ |
| DirGNN | $75.13 \pm 1.95$ | $79.74 \pm 1.40$ | $63.97 \pm 0.30$ | $73.95 \pm 0.05$ | $91.3 \pm 0.46$ |
| FaberNet | $\mathbf{76.71 \pm 1.92}$ | $\mathbf{80.33 \pm 1.19}$ | $\mathbf{64.43 \pm 0.28}$ | $\mathbf{75.10 \pm 0.03}$ | $\mathbf{92.24 \pm 0.43}$ |

As can be inferred from Table 1, **FaberNet sets new state of the art results** on all five heterophilic graph datasets above; out-performing intricate *undirected* methods specifically designed for the setting of heterophily. What is more, it also outperforms *directed spatial* methods such as Dir-GNN, whose results can be considered as reporting a best-of-three performance over multiple directed spatial methods (c.f. Appendix I or Rossi et al. (2023) for details). FaberNet also significantly out-performs MagNet. This method is a spectral model, which relies on the graph Fourier transform associated to a certain operator that is able to remain self-adjoint in the directed setting. We thus might take this gap in performance as further evidence of the utility of transcending the classical graph Fourier transform: Utilizing the holomorphic functional calculus – as opposed to the traditional graph Fourier transform – allows to base filters on (non-self-adjoint) operators more adapted to the respective task at hand. On Squirrel and Chameleon, our method performed best when using complex parameters (c.f. Table 7 in Appendix I). With MagNet being the only other method utilizing complex parameters, its performance gap to Dir-ResolvNet also implies that it is indeed the interplay of complex weights with judiciously chosen filter banks and characteristic operators that provides state-of-the-art performance; not the use of complex parameters alone.

### 5.2 DIR-RESOLVNET: DIGRAPH REGRESSION AND SCALE-INSENSITIVITY

We test the properties of our **Dir-ResolvNet** HoloNet via graph regression experiments. Weighted-directed datasets containing both node-features and graph-level targets are currently still scarce. Hence we follow Koke et al. (2023) and evaluate on the task of molecular property prediction. While neither our Dir-ResolvNet nor baselines of Table 1 are designed for this task, such molecular data still allows for fair comparisons of expressive power and stability properties of (non-specialized) graph learning methods (Hu et al., 2020a). We utilize the QM7 dataset (Rupp et al., 2012), containing graphs of 7165

organic molecules; each containing hydrogen and up to seven types of heavy atoms. Prediction target is the molecular atomization energy. While each molecule is originally represented by its Coulomb matrix $W_{ij} = Z_i \cdot Z_j/|\vec{x}_i - \vec{x}_j|$, we modify these edge-weights: Between each heavy atom and all atoms outside its respective immediate hydrogen cloud we set $W_{ij} = Z_i^{\text{outside}} \cdot (Z_j^{\text{heavy}} - 1)/|\vec{x}_i - \vec{x}_j|$. While the sole reason for this change is to make make the underlying graphs directed (enabling comparisons of *directed* methods), we might heuristically interpret it as arising from a (partial) shielding of heavy atoms by surrounding electrons (Heinz & Suter, 2004).

**Digraph-Regression:** With $W$ as directed weight-matrix and setting $y = -1$, we evaluate Dir-ResolvNet against all other *directed* methods of Table 1. Atomic charges are used as node weights ($\mu_i = Z_i$) where applicable and one-hot encodings of atomic charges $Z_i$ provide node-wise input features. As evident from Table 2, our method produces significantly lower mean-absolute-errors (MAEs) than corresponding directed baselines: **Competitors are out-performed by a factor of two and more**. We attribute this to Dir-ResolvNets ability to discard superfluous information; thus better representing overall molecular geometries.

Table 2: QM7[kcal/mol]

| | |
|---|---|
| DirGNN | $59.01_{\pm 2.54}$ |
| MagNet | $45.31_{\pm 4.24}$ |
| DiGCN | $39.95_{\pm 6.23}$ |
| Dir-ResolvNet | $\mathbf{17.12}_{\pm 0.63}$ |

**Scale-Insensitivity:** To numerically investigate the stability properties of Dir-ResolvNet that were

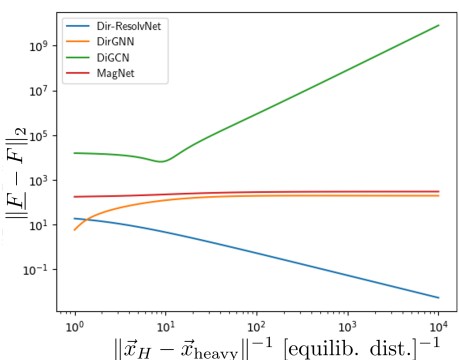

Figure 6: Feature-difference for collapsed ($\underline{F}$) and deformed ($F$) graphs.

mathematically established in Theorems 3.4 and 4.2, we translate Koke et al. (2023)'s *undirected* setup to the *directed* setting: We modify (all) molecular graphs on QM7 by deflecting hydrogen atoms (H) out of equilibrium towards the respective nearest heavy atom. This introduces a two-scale setting as in Section 3.3.2: Edge weights between heavy atoms remain the same, while weights between H-atoms and closest heavy atoms increasingly diverge. Given an original molecular graph $G$, the corresponding limit $\underline{G}$ corresponds to a coarse grained description, with heavy atoms and surrounding H-atoms aggregated into super-nodes. Feature vectors of aggregated nodes are now normalized bag-of-word vectors whose individual entries encode how much total charge of a given super-node is contributed by individual atom-types. Appendix I provides additional details.

In this setting, we compare feature vectors of collapsed graphs with feature vectors of molecules where hydrogen atoms have been deflected but have not yet arrived at the positions of nearest heavy atoms. Feature vectors are generated with the previously trained networks of Table 2. As evident from Fig. 6, **Dir-ResolvNet's feature-vectors converge** as the scale $c \sim \|\vec{x}_H - \vec{x}_{\text{heavy}}\|^{-1}$ increases; thus numerically verifying the scale-invariance Theorem 4.2. **Feature vectors of baselines do not converge**: These models are sensitive to changes in resolutions when generating graph-level features.

This difference in sensitivity is also apparent in our final experiment, where collapsed molecular graphs $\{\underline{G}\}$ are treated as a model for data obtained from a resolution-limited observation process unable to resolve individual H-atoms. Given models trained on directed higher resolution digraphs $\{G\}$, atomization energies are then to be predicted solely using coarse grained molecular digraphs. While Dir-ResolvNet's prediction accuracy remains high, performance of baselines decreases significantly if the resolution scale is reduced during inference: While Dir-ResolvNet out-performed baselines by a factor of two and higher before, this **lead increases to a factor of up to 240 if resolutions vary** (c.f. Table 3).

Table 3: QM7$^{\text{coarse}}_{\text{[kcal/mol]}}$

| | |
|---|---|
| DirGNN | $195.64_{\pm 2.20}$ |
| MagNet | $663.63_{\pm 190.358}$ |
| DiGCN | $6672.71_{\pm 2243.61}$ |
| Dir-ResolvNet | $\mathbf{27.34}_{\pm 7.55}$ |

# 6 CONCLUSION

We introduced the HoloNet framework, which allows to extend spectral networks to directed graphs. Key building blocks of these novel networks are newly indroduced holomorphic filters, no longer reliant on the graph Fourier transform. We provided a corresponding frequency perspective, investigated optimal filter-banks and discussed the interplay of filters with characteristic operators in shaping inductive biases. Experiments on real world data considered two particular HoloNet instantiations: FaberNet provided new state-of-the-art results for node classification under heterophily while Dir-ResolvNet generated feature vectors stable to resolution-scale-varying topological perturbations.

## 7 ACKNOWLEDGEMENTS

The authors thank Mariia Gladkova and Tarun Yenamandra for helpful discussions. This work was supported by the ERC Advanced Grant SIMULACRON.

## 8 REPRODUCIBILITY

We are taking great care in ensuring reproducibility of our work:

- We give complete mathematical definitions of all utilized concepts. Proofs of all Theorems are provided in the appendix (c.f. e.g. Appendices F, G and H).
- We exactly detail our newly introduced (HoloNet) framework and its two particular instances (FaberNet, Dir-ResolvNet) in the main body of our paper (c.f. Section 4).
- The experimental setups for all our experiments are described exactly in Appendix I
- All hyperparameter settings of our methods for all experiments are detailed in the appendix (c.f. Table 6, Table 7 and Section I.2).
- Our code is available at `https://github.com/ChristianKoke/HoloNets`.

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

# A   NOTATION

We provide a summary of employed notational conventions:

Table 4: Notational Conventions

| Symbol | Meaning |
| --- | --- |
| $G$ | a graph |
| $\mathcal{G}$ | Nodes of the graph $G$ |
| $N$ | number of nodes $|\mathcal{G}|$ in $G$ |
| $\underline{G}$ | Coarse grained version of graph $G$ |
| $\mu_i$ | weight of node $i$ |
| $M$ | weight matrix |
| $\langle \cdot, \cdot \rangle$ | inner product |
| $W$ | (weighted) adjacency matrix |
| $D^{\text{in/out}}$ | in/out-degree matrix |
| $L^{\text{in}}$ | in-degree graph Laplacian |
| $T$ | generic characteristic operator |
| $T^*$ | hermitian adjoint of $T$ |
| $T^\top$ | transpose of $T$ (used if and only if $T$ has only real entries) |
| $U$ | change-of-basis matrix to a (complete) basis consisting of eigenvectors (used in the undirected setting only) |
| $\sigma(T)$ | spectrum (i.e. collection of eigenvalues) of $T$ |
| $\lambda$ | an eigenvalue |
| $g$ | a holomorphic function |
| $g(T)$ | function $g$ applied to operator $T$ |
| $\Psi_i$ | an element of a filter-bank |
| $z, y$ | complex numbers |
| $U$ | subset of $\mathbb{C}$ |
| $R_z(L^{\text{in}})$ | the resolvent $(L^{\text{in}} - z \cdot Id)^{-1}$ |
| $c$ | a weight-/resolution- scale |
| $J^\downarrow, J^\uparrow$ | projection and interpolation operator |
| $\Phi$ | map associated to a graph convolution network |
| $\Omega$ | graph-level aggregation mechanism |
| $Z_i$ | atomic charge of atom corresponding to node $i$ |
| $\vec{x}_i$ | Cartesian position of atom corresponding to node $i$ |
| $\frac{Z_i Z_j}{|\vec{x}_i - \vec{x}_j|}$ | Coulomb interaction between atoms $i$ and $j$ |
| $|\vec{x}_i - \vec{x}_j|$ | Euclidean distance between $x_i$ and $x_j$ |

# B   OPERATORS BEYOND THE SELF-ADJOINT SETTING

Here we briefly recapitulate facts from linear algebra regarding self-adjoint and non-self-adjoint matrices, their spectral theory and canonical decompositions.

**Self-Adjoint Matrices:**   Let us begin with the familiar self adjoint matrices. Given a vector space $V$ with inner product $\langle \cdot, \cdot \rangle_V$ consider a linear map $T : V \to V$. The adjoint $T^*$ of the map $T$ is defined by the demanding that for all $x, y \in V$ we have

$$\langle x, Ty \rangle_V = \langle T^* x, y \rangle_V.$$

If we have $V = \mathbb{C}^d$ and the inner product is simply the standard scalar product $\langle x, y \rangle_{\mathbb{C}^d} = \overline{x}^\top y$, we have

$$T^* = \overline{T}^\top.$$

Here $\overline{x}$ denotes the complex conjugate of $x$ and $A^\top$ denotes the transpose of $A$.

An eigenvector- eigenvalue pair, is a pair of a scalar $\lambda \in \mathbb{C}$ and vector $v \in V$ so that

$$(T - \lambda \cdot Id)v = 0.$$

It is a standard result in linear algebra (see e.g. (Kato, 1976)) that the spectrum $\sigma(T)$ of all eigenvalues $\lambda$ of $T$ contains only real numbers (i.e. $\lambda \in \mathbb{R}$), if $T$ is self-adjoint. What is more, there exists a complete basis $\{v_i\}_{i=1}^d$ of such eigenvectors that span all of $V$. These eigenvectors may be chosen to satisfy

$$\langle v_i, v_j \rangle = \delta_{ij}, \tag{5}$$

with the Kronecker delta $\delta_{ij}$. As a consequence of this fact, there exists a family of (so called) spectral projections $\{P_\lambda\}_\lambda$ that have the following properties (c.f. e.g. Teschl (2014) for a proof)

- Each $P_\lambda$ is a linear map on $V$ ($P_\lambda : V \to V$).
- For all eigenvectors $v$ to the eigenvalue $\lambda$ (i.e. $v$ such that $(T - \lambda Id)v = 0$), we have
$$P_\lambda v = v.$$

- If $w$ is an eigenvector to a different eigenvalue $\mu \in \sigma(T)$ (i.e. $w$ such that $(T - \mu \cdot Id) \cdot w = 0$ and $\mu \neq \lambda$), we have
$$P_\lambda w = 0.$$

- Each $P_\lambda$ is a projection (i.e. satisfies $P_\lambda \cdot P_\lambda = P_\lambda$).
- Each $P_\lambda$ is self-adjoint (i.e. $P_\lambda = P_\lambda^*$).
- Each $P_\lambda$ commutes with $T$ (i.e. $P_\lambda \cdot T = T \cdot P_\lambda$).
- The family $\{P_\lambda\}_{\lambda \in \sigma(T)}$ form a resolution of the identity:
$$\sum_{\lambda \in \sigma(T)} P_\lambda = Id.$$

These properties together then allow us to "diagonalise" the operator $T$ and write it as a sum over its eigenvalues:
$$T = \sum_{\lambda \in \sigma(T)} \lambda \cdot P_\lambda.$$

Applying a generic function may then be defined as (Teschl, 2014)
$$g(T) := \sum_{\lambda \in \sigma(T)} g(\lambda) \cdot P_\lambda.$$

**Non self-adjoint operators:** If the operator $T$ is no longer self adjoint, eigenvalues no longer need to be in real, but are generically complex (i.e. $\lambda \in \mathbb{C}$). Furthermore, while there still exist eigenvectors these no longer need to form a basis of the space $V$. What instead becomes important in this setting, are so called *generalized eigenveectors*. A generalized eigenvector $w$ associated to the eigenvalue $\lambda$ is a vector for which we have
$$(T - \lambda \cdot Id)^n \cdot w = 0$$

for some power $n \in \mathbb{N}$. Clearly each actual *eigenvector* is also a generalized eigenvector (simply for $n = 1$). What can be shown (Kato, 1976) is that there is a basis of $V$ consisting purely of *generalized* eigenvectors (each associated to some eigenvalue $\lambda$). These now however need no longer be orthogonal (i.e. they need not satisfy (5)).

There then exists a family of spectral projections $\{P_\lambda\}_\lambda$ that have the following modified set of properties (c.f. e.g. Kato (1976) for a proof)

- Each $P_\lambda$ is a linear map on $V$ ($P_\lambda : V \to V$).
- For all generalized eigenvectors $w$ to the eigenvalue $\lambda$ (i.e. $w$ such that $(T - \lambda Id)^n \cdot w = 0$ for some $n \in \mathbb{N}$), we have
$$P_\lambda w = w.$$

- If $w$ is a generalized eigenvector to a different eigenvalue $\mu \in \sigma(T)$ (i.e. $w$ such that $(T - \mu \cdot Id)^m \cdot w = 0$ for some $m \in N$ and $\mu \neq \lambda$), we have
$$P_\lambda w = 0.$$

- Each $P_\lambda$ is a projection (i.e. satisfies $P_\lambda \cdot P_\lambda = P_\lambda$).

- Each $P_\lambda$ commutes with $T$ (i.e. $P_\lambda \cdot T = T \cdot P_\lambda$).
- The family $\{P_\lambda\}_{\lambda \in \sigma(T)}$ form a resolution of the identity:

$$\sum_{\lambda \in \sigma(T)} P_\lambda = Id.$$

Using these facts, we may thus decompose each operator $T$ into a sum as

$$T = \sum_{\lambda \in \sigma(T)} \lambda \cdot P_\lambda + \sum_{\lambda \in \sigma(T)} (T - \lambda \cdot Id) \cdot P_\lambda. \tag{6}$$

This sum decomposition is referred to as Jordan-Chevalley decomposition. Importantly, for each $\lambda \in \sigma(T)$ there is a corresponding natural number $m_\lambda \in \mathbb{N}$ referred to the *algebraic multiplicity* of the eigenvalue $\lambda$. This number $m_\lambda$ counts, how many generalized eigenvectors $\{w_i^\lambda\}_{i=1}^{m_\lambda}$ are associated to the generalized eigenvalue $\lambda$. These generalized eigenvectors can be chosen such that

$$(T - \lambda \cdot Id) \cdot w_i^\lambda = (T - \lambda \cdot Id) \cdot P_\lambda \cdot w_i^\lambda = w_{i+1}^\lambda,$$

if $i \leqslant m_\lambda - 1$. For the case $i \geqslant m_\lambda$ we have

$$(T - \lambda \cdot Id) \cdot w_{m_\lambda}^\lambda = 0.$$

In total, this implies the nilpotency-relation

$$(T - \lambda \cdot Id)^{m_\lambda} \cdot P_\lambda = [(T - \lambda \cdot Id) \cdot P_\lambda]^{m_\lambda} = 0.$$

## C SPECTRUM OF ADJACENCY OF THREE-NODE DIRECTED PATH GRAPH:

The adjacency matrix associated to the graph depicted in Figure 4 is given by

$$W = \begin{pmatrix} 0 & 0 & 0 \\ 1 & 0 & 0 \\ 0 & 1 & 0 \end{pmatrix}.$$

Denote by $e_i$ the $i^{\text{th}}$ canonical basis vector. Then

$$W \cdot e_1 = e_2, \quad W \cdot e_2 = e_3, \quad W \cdot e_3 = 0.$$

Clearly $e_3$ is an eigenvector to the eigenvalue $\lambda = 0$, so that $0 \in \sigma(W)$. Suppose there exists an eigenvector $v$ associated to a different eigenvalue $\mu \neq 0$:

$$W \cdot v = \mu \cdot v.$$

Then we have

$$W^3 \cdot v = \mu^3 \cdot v$$

Since $\{e_1, e_2, 2_3\}$ clearly is a basis, there are numbers $a, b, c$ so that

$$v = ae_1 + be_2 + ce_3.$$

But then

$$\mu^3 \cdot v = W^3 \cdot v = aW^3 \cdot e_1 + bW^3 \cdot e_2 + cW^3 \cdot e_3 = a \cdot 0 + b \cdot 0 + c \cdot 0.$$

Thus we have $\mu = 0$ and hence zero is indeed the only eigenvalue.

## D COMPLEX DIFFERENTIABILITY

Complete introductions into this subject may be found in Ahlfors (1966) or (Bak & Newman, 2017).

For a complex valued function $f : \mathbb{C} \to \mathbb{C}$ of a single complex variable, the derivative of $f$ at a point $z_0 \in \mathbb{C}$ in its domain of definition is defined as the limit

$$f'(z_0) := \lim_{z \to z_0} \frac{f(z) - f(z_0)}{z - z_0}.$$

For this limit to exist, it needs to be independent of the 'direction' in which $z$ approaches $z_0$, which is a stronger requirement than being real-differentiable.

A function is called holomorphic on an open set $U$ if it is complex differentiable at every point in $U$. The value $g(\lambda)$ of any such function $g$ at $\lambda$ may be reproduced as

$$g(\lambda) = -\frac{1}{2\pi i} \oint_S \frac{g(z)}{\lambda - z} dz \tag{7}$$

for any circle $S \subseteq \mathbb{C}$ encircling $\lambda$ so that $S$ is completely contained within $U$ and may be contracted to a point without leaving $U$. This equation is referred to as Cauchy's integral formula (Bak & Newman, 2017).

In fact, the integration contour need not be a circle $S$, but may be the boundary of any so called Cauchy domain containing $\lambda$:

**Definition D.1.** A subset $D$ of the complex plane $\mathbb{C}$ is called a Cauchy domain if $D$ is open, has a finite number of components (the closure of two of which are disjoint) and the boundary of $\partial D$ of $D$ is composed of a finite number of closed rectifiable Jordan curves, no two of which intersect.

Integrating around any such boundary then reproduces the value of $g$ at $\lambda$.

# E    ADDITIONAL DETAILS ON THE HOLOMORPHIC FUNCTIONAL CALCULUS

**Fundamental Definition:**    In order to define the matrix $g(T)$, the formal replacement $\lambda \mapsto T$ is made on both sides of the Cauchy formula (7), with the path $\Gamma$ now not only encircling a single value $\lambda$ but all eigenvalues $\lambda \in \sigma(T)$ (c.f. also Fig. 7):

$$g(T) := -\frac{1}{2\pi i} \oint_\Gamma g(z) \cdot (T - z \cdot Id)^{-1} dz \tag{8}$$

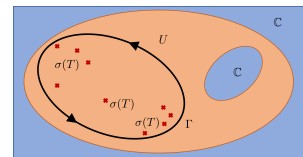

The integral is well defined since all eigenvalues of $T$ are assumed to lie *inside* the path $\Gamma$. For any choice of integration variable $z$ *on* this path $\Gamma$, the matrix $(T - z \cdot Id)$ is thus indeed invertible, since $z$ is never an eigenvalue.

Figure 7: Operator Integral (8)

This integral is also known as Dunford-Taylor integral, and the holomorphic functional calculus is also sometimes called Riesz–Dunford functional calculus (Kato, 1976; Post, 2012). As with the scalar valued integral (7), it can be shown that the precise path $\Gamma$ is not important, as long as it encircles all eigenvalues counter-clockwise and is contractable to a single point within the domain of definition $U$ for the function $g$.

**Spectral characterization of the Holomorphic Functional Calculus:**    As stated in Section 3.2, the operator $g(T)$ may also be characterized spectrally. Writing $T = \sum_{\lambda \in \sigma(T)} \lambda \cdot P_\lambda + \sum_{\lambda \in \sigma(T)} (T - \lambda \cdot Id) \cdot P_\lambda$ as in (6), it can then be shown (Kato, 1976), that the spectral action of a given function $g$ is given as

$$g(T) = \sum_{\lambda \in \sigma(T)} g(\lambda) P_\lambda + \sum_{\lambda \in \sigma(T)} \left[ \sum_{n=1}^{m_\lambda - 1} \frac{g^{(n)}(\lambda)}{n!} (T - \lambda \cdot Id)^n \right] P_\lambda. \tag{9}$$

A proof of this can be found in Chapter 1 of Kato (1976).

One of the strong properties of complex differentiability previously alluded to is, that as soon as a function is complex-differentiable once, it is already complex differentiable infinitely often (Bak & Newman, 2017). Hence the $n^{\text{th}}$-derivative in (9) does indeed exist.

With the preparations of Appendix B, we can interpret the sum

$$S = \sum_{\lambda \in \sigma(T)} \left[ \sum_{n=1}^{m_\lambda - 1} \frac{g^{(n)}(\lambda)}{n!} (T - \lambda \cdot Id)^n \right] P_\lambda \tag{10}$$

further:

We first note, that summing up to $n \geqslant m_\lambda$ would not make sense, since the nilpotency relation

$$(T - \lambda \cdot Id)^{m_\lambda} \cdot P_\lambda = [(T - \lambda \cdot Id) \cdot P_\lambda]^{m_\lambda} = 0.$$

tells us that any such term would be zero.

Furthermore, the factor $(T - \lambda \cdot Id)^k$ can be considered to be a "ladder operator" acting on a basis $\{w_i^\lambda\}\}_{i=1}^{m_\lambda}$ of generalized eigenvectors spanning the generalized eigenspace associated to the eigenvalue $\lambda$: It acts as

$$(T - \lambda \cdot Id) \cdot P_\lambda \cdot w_i^\lambda = w_{i+1}^\lambda,$$

as discussed in Appendix B.

The values $g^{(n)}(\lambda)$ in (10) can then be interpreted as weighing the individual "permutations" $(T - \lambda \cdot Id)^n P_\lambda$ of basis elements in the generalized eigenspace associated to the eigenvalue $\lambda$.

Finally we note that the spectrum of the new operator $g(T)$ is given as

$$\sigma(g(T)) = g(\sigma(T)) := \{g(\lambda) : \lambda \in \sigma(T)\}.$$

This is known as the "Spectral Mapping Theorem" (Kato, 1976).

**Compatibility with Algebraic relations**   Here we prove compatibility of the holomorphic functional calculus with algebraic relations. For ease in writing, we will use the notation

$$(z \cdot Id - T)^{-1} \equiv \frac{1}{z - T}$$

below.

Let us begin with monomials:

**Lemma E.1.** Applying the function $g(\lambda) = \lambda^k$ to $T$ yields $T^k$.

*Proof.* We want to prove that

$$\frac{1}{2\pi i} \oint_\Gamma \frac{z^k}{z - T} dz = T^k$$

To this end, we use the Neumann series characterisation of the resolvent (Teschl, 2014)

$$(z - T)^{-1} = \frac{1}{z} \sum_{n=0}^\infty \left(\frac{T}{z}\right)^n,$$

which is valid for $|z| > \|T\|$. Substituting with $g(\lambda) = \lambda^k$ yields

$$g(T) = \frac{1}{2\pi i} \oint_\Gamma \left(\sum_{n=0}^\infty \frac{T^n}{z^{n+1-k}}\right) dz$$

which we may rewrite as

$$\sum_{n=0}^\infty \left(\frac{1}{2\pi i} \oint_\Gamma \frac{dz}{z^{n+1-k}}\right) = \sum_{n=0}^\infty T^n \cdot \delta_{nk} = T^k$$

Here we used the relation (c.f. e.g. Bak & Newman (2017))

$$\frac{1}{2\pi i} \oint_\Gamma \frac{dz}{z^{n+1-k}} = \delta_{nk}.$$

$\square$

Next we prove that the holomorphic functional calculus is also consistent with inversion:

**Lemma E.2.** If $y$ is not an eigenvalue of $T$, applying the function $g(\lambda) = \left(\frac{1}{\lambda - y}\right)^k$ to $T$ yields

$$g(T) = [(T - y \cdot Id)^{-1}]^k.$$

For ease in notation we will write $[(T - y \cdot Id)^{-1}]^k \equiv (T - y \cdot Id)^{-k}$.

*Proof.* What want to prove, is thus the equality

$$(y \cdot Id - T)^{-k} := \frac{1}{2\pi i} \oint_\Gamma (y - z)^{-k} \cdot (zId - T)^{-1} dz,$$

We first note that for the resolvent $R_y(T) = (T - y \cdot Id)^{-1}$ we may write

$$R_x(T) = \sum_{n=0}^{\infty} (x - y)^n (-1)^n R_y(t)^{n+1}$$

for $|x - y| \leqslant \|R_y(T)\|$ using standard results in matrix analysis (namely the 'Neumann Characterisation of the Resolvent' which is obtained by repeated application of a resolvent identity; c.f. Post (2012) or Teschl (2014) for more details). We thus find

$$\frac{1}{2\pi i} \oint_\Gamma \left(\frac{1}{y - z}\right)^k \frac{1}{zId - T} dz = \frac{1}{2\pi i} \oint_\Gamma \left(\frac{1}{y - z}\right)^k \sum_{n=0}^{\infty} (y - z)^n R_y(T)^{n+1}.$$

Using the fact that

$$\frac{1}{2\pi i} \oint_\Gamma (z - y)^{n-k-1} dz = \delta_{nk}$$

then yields the claim. $\qquad \square$

## F    PROOF OF THEOREM 3.3

To increase readability, we here use the notation

$$\Delta \equiv L^{\text{in}}.$$

In this section, we then prove Theorem 3.3. For convenience, we first restate the result – together with the definitions leading up to it – again:

**Definition F.1.** Denote by $\underline{\mathcal{G}}$ the set of reaches in $G_{\text{high}}$. We give this set a graph structure as follows: Let $R$ and $P$ be elements of $\underline{\mathcal{G}}$ (i.e. reaches in $G_{\text{high}}$). We define the real number

$$\underline{W}_{RP} = \sum_{r \in R} \sum_{p \in P} W_{rp},$$

with $r$ and $p$ nodes in the original graph $G$. We define the set of edges $\underline{\mathcal{E}}$ on $\underline{G}$ as

$$\underline{\mathcal{E}} = \{(R, P) \in \underline{\mathcal{G}} \times \underline{\mathcal{G}} : \underline{W}_{RP} > 0\}$$

and assign $\underline{W}_{RP}$ as weight to such edges. Node weights of limit nodes are defined similarly as aggregated weights of all nodes $r$ (in $G$) contained in the reach $R$ as

$$\underline{\mu}_R = \sum_{r \in R} \mu_r.$$

In order to translate signals between the original graph $G$ and the limit description $\underline{G}$, we need translation operators mapping signals from one graph to the other:

**Definition F.2.** Denote by $\mathbb{1}_R$ the vector that has 1 as entries on nodes $r$ belonging to the connected (in $G_{\text{hign}}$) component $R$ and has entry zero for all nodes not in $R$. We define the down-projection operator $J^{\downarrow}$ component-wise via evaluating at node $R$ in $\underline{\mathcal{G}}$ as

$$(J^{\downarrow}x)_R = \langle \mathbb{1}_R, x \rangle / \underline{\mu}_R.$$

The upsampling operator $J^{\uparrow}$ is defined as

$$J^{\uparrow}u = \sum_R u_R \cdot \mathbb{1}_R; \tag{11}$$

where $u_R$ is a scalar value (the component entry of $u$ at $R \in \underline{\mathcal{G}}$) and the sum is taken over all reaches in $G_{\text{high}}$.

The result we then have to prove is the following:

**Theorem F.3.** We have $R_z(\Delta) \to J^{\uparrow}R_z(\underline{\Delta})J^{\downarrow}$ as the weight scale $c$ increases. Explicitly,

$$\left\| R_z(\Delta) - J^{\uparrow}R_z(\underline{\Delta})J^{\downarrow} \right\| \to 0 \text{ as } c \longrightarrow \infty$$

holds.

The proof closely follows that of the corresponding result in (Koke et al., 2023).

*Proof.* We will split the proof of this result into multiple steps. For $z < 0$ Let us denote by

$$R_z(\Delta) = (\Delta - zId)^{-1},$$
$$R_z(\Delta_{high}) = (\Delta_{high} - zId)^{-1}$$
$$R_z(\Delta_{regular}) = (\Delta_{regular} - zId)^{-1}$$

the resolvents corresponding to $\Delta$, $\Delta_{high}$ and $\Delta_{regular}$ respectively.
Our first goal is establishing that we may write

$$R_z(\Delta) = [Id + R_z(\Delta_{high})\Delta_{regular}]^{-1} \cdot R_z(\Delta_{high})$$

This will follow as a consequence of what is called the second resolvent formula (Teschl, 2014):

"Given operators $A, B$, we may write

$$R_z(A + B) - R_z(A) = -R_z(A)BR_z(A + B)."$$

In our case, this translates to

$$R_z(\Delta) - R_z(\Delta_{high}) = -R_z(\Delta_{high})\Delta_{\text{regular}}R_z(\Delta)$$

or equivalently

$$[Id + R_z(\Delta_{high})\Delta_{\text{regular}}] R_z(\Delta) = R_z(\Delta_{high}).$$

Multiplying with $[Id + R_z(\Delta_{high})\Delta_{\text{regular}}]^{-1}$ from the left then yields

$$R_z(\Delta) = [Id + R_z(\Delta_{high})\Delta_{regular}]^{-1} \cdot R_z(\Delta_{high})$$

as desired.
Hence we need to establish that $[Id + R_z(\Delta_{high})\Delta_{regular}]$ is invertible for $z < 0$.

To establish a contradiction, assume it is not invertible. Then there is a signal $x$ such that

$$[Id + R_z(\Delta_{high})\Delta_{regular}] x = 0.$$

Multiplying with $(\Delta_{\text{high}} - zId)$ from the left yields

$$(\Delta_{\text{high}} + \Delta_{\text{regular}} - zId)x = 0$$

which is precisely to say that

$$(\Delta - zId)x = 0$$

But since $\Delta$ is a graph Laplacian, it only has eigenvalues with non-negative real part (Veerman & Lyons, 2020). Hence we have reached our contradiction and established

$$R_z(\Delta) = [Id + R_z(\Delta_{high})\Delta_{regular}]^{-1} R_z(\Delta_{high}).$$

Our next step is to establish that

$$R_z(\Delta_{high}) \to \frac{P_0^{high}}{-z},$$

where $P_0^{high}$ is the spectral projection onto the eigenspace corresponding to the lowest lying eigenvalue $\lambda_0(\Delta_{high}) = 0$ of $\Delta_{high}$.

Indeed, using the spectral characterization of the holomorphic functional calculus, we may write

$$g(\Delta_{high}) = \sum_{\lambda \in \sigma(\Delta_{high})} g(\lambda)P_\lambda + \sum_{\lambda \in \sigma(\Delta_{high})} \left[ \sum_{n=1}^{m_\lambda - 1} \frac{g^{(n)}(\lambda)}{n!} (\Delta_{high} - \lambda \cdot Id)^n \right] P_\lambda.$$

with

$$g(\lambda) = \frac{1}{\lambda - z}.$$

Scaling the operator $\Delta_{high}$ as

$$\Delta_{high} \longmapsto c \cdot \Delta_{high}$$

also scales all corresponding eigenvalues $\lambda$ as $\lambda \mapsto c \cdot \lambda$, while leaving the spectral projections $P_\lambda$ invariant (Kato, 1976). Thus taking the limit $c \to \infty$ we indeed find

$$\lim_{c \to \infty} g(c \cdot \Delta_{high}) = \frac{P_0^{high}}{-z}.$$

Our next task is to use this result in order to show that the difference

$$I := \left\| \left[ Id + \frac{P_0^{high}}{-z} \Delta_{regular} \right]^{-1} \frac{P_0^{high}}{-z} - [Id + R_z(\Delta_{high})\Delta_{regular}]^{-1} R_z(\Delta_{high}) \right\|$$

goes to zero as $c \to \infty$.

To this end we first note that the relation

$$[A + B - zId]^{-1} = [Id + R_z(A)B]^{-1} R_z(A)$$

provided to us by the second resolvent formula, implies

$$[Id + R_z(A)B]^{-1} = Id - B[A + B - zId]^{-1}.$$

Thus we have

$$\left\| [Id + R_z(\Delta_{high})\Delta_{regular}]^{-1} \right\| \leqslant 1 + \|\Delta_{regular}\| \cdot \|R_z(\Delta)\|$$

$$\leqslant 1 + \frac{\|\Delta_{regular}\|}{|z|}.$$

With this, we have

$$\left\|\left[Id + \frac{P_0^{high}}{-z}\Delta_{regular}\right]^{-1} \cdot \frac{P_0^{high}}{-z} - R_z(\Delta)\right\|$$

$$= \left\|\left[Id + \frac{P_0^{high}}{-z}\Delta_{regular}\right]^{-1} \cdot \frac{P_0^{high}}{-z} - [Id + R_z(\Delta_{high})\Delta_{regular}]^{-1} \cdot R_z(\Delta_{high})\right\|$$

$$\leqslant \left\|\frac{P_0^{high}}{-z}\right\| \cdot \left\|\left[Id + \frac{P_0^{high}}{-z}\Delta_{regular}\right]^{-1} - [Id + R_z(\Delta_{high})\Delta_{regular}]^{-1}\right\|$$

$$+ \left\|\frac{P_0^{high}}{-z} - R_z(\Delta_{\text{high}})\right\| \cdot \left\|[Id + R_z(\Delta_{high})\Delta_{regular}]^{-1}\right\|$$

$$\leqslant \frac{1}{|z|}\left\|\left[Id + \frac{P_0^{high}}{-z}\Delta_{regular}\right]^{-1} - [Id + R_z(\Delta_{high})\Delta_{regular}]^{-1}\right\| + \epsilon$$

Hence it remains to bound the left hand summand. For this we use the following fact (c.f. Horn & Johnson (2012), Section 5.8. "Condition numbers: inverses and linear systems"):

Given square matrices $A, B, C$ with $C = B - A$ and $\|A^{-1}C\| < 1$, we have

$$\|A^{-1} - B^{-1}\| \leqslant \frac{\|A^{-1}\| \cdot \|A^{-1}C\|}{1 - \|A^{-1}C\|}.$$

In our case, this yields (together with $\|P_0^{high}\| = 1$) that

$$\left\|\left[Id + P_0^{high}/(-z) \cdot \Delta_{regular}\right]^{-1} - [Id + R_z(\Delta_{high})\Delta_{regular}]^{-1}\right\|$$

$$\leqslant \frac{(1 + \|\Delta_{\text{regular}}\|/|z|)^2 \cdot \|\Delta_{\text{regular}}\| \cdot \|\frac{P_0^{\text{high}}}{-z} - R_z(\Delta_{\text{high}})\|}{1 - (1 + \|\Delta_{\text{regular}}\|/|z|) \cdot \|\Delta_{\text{regular}}\| \cdot \|\frac{P_0^{\text{high}}}{-z} - R_z(\Delta_{\text{high}})\|}$$

For $c$ sufficiently large, we have

$$\| - P_0^{\text{high}}/z - R_z(\Delta_{\text{high}})\| \leqslant \frac{1}{2\left(1 + \|\Delta_{\text{regular}}\|/|z|\right)}$$

so that we may estimate

$$\left\|\left[Id + \Delta_{regular}\frac{P_0^{high}}{-z}\right]^{-1} - [Id + \Delta_{regular}R_z(\Delta_{high})]^{-1}\right\|$$

$$\leqslant 2 \cdot (1 + \|\Delta_{\text{regular}}\|) \cdot \|\frac{P_0^{\text{high}}}{-z} - R_z(\Delta_{\text{high}})\|$$

$$\to 0$$

Thus we have now established

$$\left|\left[Id + \frac{P_0^{high}}{-z}\Delta_{regular}\right]^{-1} \cdot \frac{P_0^{high}}{-z} - R_z(\Delta)\right| \longrightarrow 0.$$

Hence we are done with the proof, as soon as we can establish

$$\left[-zId + P_0^{high}\Delta_{regular}\right]^{-1} P_0^{high} = J^{\uparrow}R_z(\underline{\Delta})J^{\downarrow},$$

with $J^\uparrow, \underline{\Delta}, J^\downarrow$ as defined above. To this end, we first note that since the left-kernel and right-kernel of $\Delta_{\text{high}}$ are the same (since in-degrees are the same as out degrees), we have

$$J^\uparrow \cdot J^\downarrow = P_0^{high} \tag{12}$$

and

$$J^\downarrow \cdot J^\uparrow = Id_{\underline{G}}. \tag{13}$$

Indeed, the relation (12) follows from the fact that the eigenspace corresponding to the eignvalue zero is spanned by the vectors $\{\mathbb{1}_R\}_R$, with $\{R\}$ the reaches of $G_{\text{high}}$ (c.f. Veerman & Lyons (2020)). Equation (13) follows from the fact that

$$\langle \mathbb{1}_R, \mathbb{1}_R \rangle = \underline{\mu}_R.$$

With this we have

$$\left[ Id + P_0^{high} \Delta_{regular} \right]^{-1} P_0^{high} = \left[ Id + J^\uparrow J^\downarrow \Delta_{regular} \right]^{-1} J^\uparrow J^\downarrow.$$

To proceed, set

$$\underline{x} := F^\downarrow x$$

and

$$\mathscr{X} = \left[ P_0^{high} \Delta_{regular} - zId \right]^{-1} P_0^{high} x.$$

Then

$$\left[ P_0^{high} \Delta_{regular} - zId \right] \mathscr{X} = P_0^{high} x$$

and hence $\mathscr{X} \in \text{Ran}(P_0^{high})$. Thus we have

$$J^\uparrow J^\downarrow (\Delta_{\text{regular}} - zId) J^\uparrow J^\downarrow \mathscr{X} = J^\uparrow J^\downarrow x.$$

Multiplying with $J^\downarrow$ from the left yields

$$J^\downarrow (\Delta_{\text{regular}} - zId) J^\uparrow J^\downarrow \mathscr{X} = J^\downarrow x.$$

Thus we have

$$(J^\downarrow \Delta_{\text{regular}} J^\uparrow - zId) J^\uparrow J^\downarrow \mathscr{X} = J^\downarrow x.$$

This – in turn – implies

$$J^\uparrow J^\downarrow \mathscr{X} = \left[ J^\downarrow \Delta_{\text{regular}} J^\uparrow - zId \right]^{-1} J^\downarrow x.$$

Using

$$P_0^{high} \mathscr{X} = \mathscr{X},$$

we then have

$$\mathscr{X} = J^\uparrow \left[ J^\downarrow \Delta_{\text{regular}} J^\uparrow - zId \right]^{-1} J^\downarrow x.$$

We have thus concluded the proof if we can prove that $J^\downarrow \Delta_{\text{regular}} J^\uparrow$ is the Laplacian corresponding to the graph $\underline{G}$ defined in Definition F.1. But this is a straightforward calculation. □

As a corollary, we find

**Corollary F.4.** We have

$$R_z(\Delta)^k \to J^\uparrow R^k(\underline{\Delta}) J^\downarrow$$

*Proof.* This follows directly from the fact that

$$J^\downarrow J^\uparrow = Id_{\underline{G}}.$$

□

This thus establishes Theorem 3.4.

## G    STABILITY UNDER SCALE VARIATIONS

Here we provide details on the scale-invariance results discussed in Section 4; most notably Theorem 4.2.

In preparation, we will first need to prove a lemma relating powers of resolvents on the original graph $G$ and its limit-description $\underline{G}$:

**Lemma G.1.** Let $\underline{R}_z := (\underline{\Delta} - zId)^{-1}$ and $R_z := (\Delta - zId)^{-1}$. For any natural number $k$, we have

$$\|J^\uparrow \underline{R}_z^k J^\downarrow - R_z^k\| \leqslant k \cdot A^{k-1} \|J^\uparrow \underline{R}_z J^\downarrow - R_z\|$$

for

$$\|R_z(\Delta)\|, \|R_z(\underline{\Delta})\| \leqslant A$$

*Proof.* We note that for arbitrary matrices $T, \widetilde{T}$, we have

$$\widetilde{T}^k - T^k = \widetilde{T}^{k-1}(\widetilde{T} - T) + (\widetilde{T}^{k-1} - T^{k-1})T$$
$$= \widetilde{T}^{k-1}(\widetilde{T} - T) + \widetilde{T}^{k-2}(\widetilde{T} - T)T + (\widetilde{T}^{k-2} - T^{k-2})T^2.$$

Iterating this, using the fact that $\|R_z(\Delta)\|$ stays bounded as $c \to \infty$, since

$$\|R_z(\Delta)\| \to \|J^\uparrow R_z(\underline{\Delta})J^\downarrow\| \leqslant A$$

for some constant $A$ together with $\|J^\uparrow\|, \|J^\downarrow\| \leqslant 1$ and

$$J^\uparrow \underline{R}_z^k J^\downarrow = \left(J^\uparrow \underline{R}_z J^\downarrow\right)^k$$

(which holds since $J^\downarrow J^\uparrow = Id_{\underline{G}}$) then yields the claim.

$\square$

Hence let us now prove a node-level stability result:

**Theorem G.2.** Let $\Phi_L$ and $\underline{\Phi}_L$ be the maps associated to Dir-ResolvNets with the same learned weight matrices and biases but deployed on graphs $G$ and $\underline{G}$ as defined in Section 3.3.2. We have

$$\|\Phi_L(X) - J^\uparrow \underline{\Phi}_L(J^\downarrow X)\|_2 \leqslant (C_1(\mathscr{W}, A) \cdot \|X\|_2 + C_2(\mathscr{W}, \mathscr{B}, A)) \cdot \left\|R_z(\Delta) - J^\uparrow R_z(\underline{\Delta})J^\downarrow\right\| \quad (14)$$

with $A$ a constant such that

$$\|R_z(\Delta)\|, \|R_z(\underline{\Delta})\| \leqslant A.$$

*Proof.* Let us define

$$\underline{X} := J^\downarrow X.$$

Let us further use the notation $\underline{R}_z := (\underline{\Delta} - zId)^{-1}$ and $R_z := (\Delta - zId)^{-1}$.

Denote by $X^\ell$ and $\widetilde{X}^\ell$ the (hidden) feature matrices generated in layer $\ell$ for networks based on resolvents $R_z$ and $\underline{R}_z$ respectively: I.e. we have

$$X^\ell = \rho \left( \sum_{k=1}^K R_z^k X^{\ell-1} W_k + B^\ell \right)$$

and

$$\widetilde{X}^\ell = \rho \left( \sum_{k=1}^K \underline{R}_z^k \widetilde{X}^{\ell-1} W_k + \underline{B}^\ell \right).$$

Here, since bias terms are proportional to constant vectors on the graphs, we have

$$J^\downarrow B = \underline{B}$$

and

$$J^\uparrow \underline{B} = B \quad (15)$$

for bias matrices $B$ and $\underline{B}$ in networks deployed on $G$ and $\underline{G}$ respectively.

We then have

$$
\begin{aligned}
&\|\Phi_L(X) - J^\uparrow \underline{\Phi}_L(J^\downarrow X)\| \\
=& \|X^L - J^\uparrow \widetilde{X}^L\| \\
=& \left\| \rho\left( \sum_{k=1}^{K} R_z^k X^{L-1} W_k^L + B^L \right) - J^\uparrow \rho\left( \sum_{k=1}^{K} \underline{R}_z^k \widetilde{X}^{L-1} W_k^L + \underline{B}^L \right) \right\| \\
=& \left\| \rho\left( \sum_{k=1}^{K} R_z^k X^{L-1} W_k^L + B^L \right) - \rho\left( \sum_{k=1}^{K} J^\uparrow \underline{R}_z^k \widetilde{X}^{L-1} W_k^L + B^L \right) \right\|.
\end{aligned}
$$

Here we used the fact that since $\rho(\cdot)$ maps positive entries to positive entries and acts pointwise, it commutes with $J^\uparrow$. We also made use of (15).
Using the fact that $\rho(\cdot)$ is Lipschitz-continuous with Lipschitz constant $D = 1$, we can establish

$$
\|\Phi_L(X) - J^\uparrow \underline{\Phi}_L(J^\downarrow X)\| \leqslant \left\| \sum_{k=1}^{K} R_z^k X^{L-1} W_k^L - \sum_{k=1}^{K} J^\uparrow \underline{R}_z^k \widetilde{X}^{L-1} W_k^L \right\|.
$$

Using the fact that $J^\downarrow J^\uparrow = Id_{\underline{G}}$, we have

$$
\|\Phi_L(X) - J^\uparrow \underline{\Phi}_L(J^\downarrow X)\| \leqslant \left\| \sum_{k=1}^{K} R_z^k X^{L-1} W_k^L - \sum_{k=1}^{K} (J^\uparrow \underline{R}_z^k J^\downarrow) J^\uparrow \widetilde{X}^{L-1} W_k^L \right\|.
$$

From this, we find (using $\|J^\uparrow\|, \|J^\downarrow\| \leqslant 1$), that

$$
\begin{aligned}
&\|X^L - J^\uparrow \widetilde{X}^L\| \\
\leqslant& \left\| \sum_{k=0}^{K} R_z^k X^{L-1} W_k^L - \sum_{k=1}^{K} (J^\uparrow \underline{R}_z^k J^\downarrow) J^\uparrow \widetilde{X}^{L-1} W_k^L \right\| \\
\leqslant& \left\| \sum_{k=1}^{K} (R_z^k - (J^\uparrow \underline{R}_z^k J^\downarrow)) X^{L-1} W_k^L \right\| + \sum_{k=1}^{K} \|J^\uparrow \underline{R}_z J^\downarrow\| \cdot \|J^\uparrow \widetilde{X}^{L-1} - X^{L-1}\| \cdot \|W_k^L\| \\
\leqslant& \left\| \sum_{k=1}^{K} (R_z^k - (J^\uparrow \underline{R}_z^k J^\downarrow)) X^{L-1} W_k^L \right\| + \|\mathscr{W}^L\|_z \cdot \|J^\uparrow \widetilde{X}^{L-1} - X^{L-1}\| \\
\leqslant& \sum_{k=1}^{K} \left\| R_z^k - (J^\uparrow \underline{R}_z^k J^\downarrow) \right\| \cdot \|X^{L-1}\| \cdot \|W_k^L\| + \|\mathscr{W}^L\|_z \cdot \|J^\uparrow \widetilde{X}^{L-1} - X^{L-1}\|
\end{aligned}
$$

Applying Lemma G.1 yields

$$
\begin{aligned}
&\|X^L - J^\uparrow \widetilde{X}^L\| \\
\leqslant& \left( \sum_{k=1}^{K} (k \cdot A^{k-1}) \|W_k^L\| \right) \cdot \|R_z - (J^\uparrow \underline{R}_z J^\downarrow)\| \cdot \|X^{L-1}\| + \|\mathscr{W}^L\|_z \cdot \|J^\uparrow \widetilde{X}^{L-1} - X^{L-1}\|.
\end{aligned}
$$

Similarly, one may establish that we have

$$
\|X^L\| \leqslant C(A) \cdot \left( \|B^L\| + \sum_{m=0}^{L} \left( \prod_{j=0}^{m} \|\mathscr{W}^{L-1-k}\|_z \right) \|B^{L-1-k}\| + \left( \prod_{\ell=1}^{L} \|\mathscr{W}^\ell\|_z \right) \cdot \|X\| \right). \quad (16)
$$

Hence the summand on the left-hand-side can be bounded in terms of a polynomial in singular values of bias- and weight matrices, as well as $\|X\|$, $A$ and most importantly the factor $\|R_z - (J^\uparrow \underline{R}_z J^\downarrow)\|$

which tends to zero.

For the summand on the right-hand-side, we can iterate the above procedure (aggregating terms like (16) multiplied by $\|R_z - (J^\uparrow \underline{R}_z J^\downarrow)\|$) until reaching the last layer $L = 1$. There we observe

$$\|X^1 - J^\uparrow \widetilde{X}^1\|$$

$$= \left\| \rho \left( \sum_{k=1}^{K} R_z^k X W_k^1 + B^1 \right) - J^\uparrow \rho \left( \sum_{k=1}^{K} \underline{R}_z^k J^\downarrow X W_k^1 + \underline{B}^1 \right) \right\|$$

$$\leqslant \left\| \sum_{k=1}^{K} R_z^k X W_k^1 - \sum_{k=1}^{K} J^\uparrow \underline{R}_z^k J^\downarrow X W_k^1 \right\|$$

$$\leqslant \left\| \sum_{k=1}^{K} (R_z^k - J^\uparrow \underline{R}_z^k J^\downarrow) X W_k^1 \right\|$$

$$\leqslant \left( \sum_{k=1}^{K} (k \cdot A^{k-1}) \|W_k^1\| \right) \cdot \|R_z - (J^\uparrow \underline{R}_z J^\downarrow)\| \cdot \|X\|$$

The last step is only possible because we let the sums over powers of resolvents start at $a = 1$ as opposed to $a = 0$. In the latter case, there would have remained a term $\|X - J^\uparrow J^\downarrow X\|$, which would not decay as $c \to \infty$.

Aggregating terms, we build up the polynomial stability constants of (14) layer by layer, and complete the proof.

$\square$

Next we transfer the previous result to the graph level setting:

**Theorem G.3.** Denote by $\Omega$ the aggregation method introduced in Section 4. With $\mu(G) = \sum_{i=1}^{N} \mu_i$ the total weight of the graph $G$, we have in the setting of Theorem G.2, that

$$\|\Omega \left( \Phi_L(X) \right) - \Omega \left( \underline{\Phi}_L(J^\downarrow X) \right) \|_2$$
$$\leqslant \sqrt{\mu(G)} \cdot (C_1(\mathscr{W}, A) \cdot \|X\|_2 + C_2(\mathscr{W}, \mathscr{B}, A)) \cdot \|R_z(\Delta) - J^\uparrow R_z(\underline{\Delta}) J^\downarrow\|.$$

*Proof.* Let us first recall that our aggregation scheme $\Omega$ mapped a feature matrix $X \in \mathbb{R}^{N \times F}$ to a graph-level feature vector $\Omega(X) \in \mathbb{R}^F$ defined component-wise as

$$\Omega(X)_j = \sum_{i=1}^{N} |X_{ij}| \cdot \mu_i.$$

In light of Theorem G.2, we are done with the proof, once we have established that

$$\|\Omega \left( \Phi_L(X) \right) - \Omega \left( \underline{\Phi}_L(J^\downarrow X) \right) \|_2 \leqslant \sqrt{\mu(G)} \cdot \|\Phi_L(X) - J^\uparrow \underline{\Phi}_L(J^\downarrow X)\|_2.$$

To this end, we first note that

$$\Omega(J^\uparrow \underline{X}) = \Omega(\underline{X}).$$

Indeed, this follows from the fact that given a reach $R$ in $G_{\text{high}}$, the map $J^\uparrow$ assigns the same feature vector to each node $r \in R \subseteq G$ (c.f. (11)), together with the fact that

$$\underline{\mu}_R = \sum_{r \in R} \mu_r.$$

Thus we have

$$\|\Omega \left( \Phi_L(X) \right) - \Omega \left( \underline{\Phi}_L(J^\downarrow X) \right) \|_2 = \|\Omega \left( \Phi_L(X) \right) - \Omega \left( J^\uparrow \underline{\Phi}_L(J^\downarrow X) \right) \|_2.$$

Next let us simplify notation and write

$$\mathcal{A} = \Phi_L(X)$$

and

$$\mathcal{B} = J^\uparrow \underline{\Phi}_L(J^\downarrow X)$$

with $\mathcal{A}, \mathcal{B} \in \mathbb{R}^{N \times F}$. We note:

$$\|\Omega\left(\Phi_L(X)\right) - \Omega\left(J^\uparrow \underline{\Phi}_L(J^\downarrow X)\right)\|_2^2 = \sum_{j=1}^{F} \left(\sum_{i=1}^{N} (|\mathcal{A}_{ij}| - |\mathcal{B}_{ij}|) \cdot \mu_i\right)^2 .$$

By means of the Cauchy-Schwarz inequality together with the inverse triangle-inequality, we have

$$\sum_{j=1}^{F} \left(\sum_{i=1}^{N} (|\mathcal{A}_{ij}| - |\mathcal{B}_{ij}|) \cdot \mu_i\right)^2 \leqslant \sum_{j=1}^{F} \left[\left(\sum_{i=1}^{N} |\mathcal{A}_{ij} - \mathcal{B}_{ij}|^2 \cdot \mu_i\right) \cdot \left(\sum_{i=1}^{N} \mu_i\right)\right]$$

$$= \sum_{j=1}^{F} \left(\sum_{i=1}^{N} |\mathcal{A}_{ij} - \mathcal{B}_{ij}|^2 \cdot \mu_i\right) \cdot \mu(G).$$

Since we have

$$\|\Phi_L(X) - J^\uparrow \underline{\Phi}_L(J^\downarrow X)\|_2^2 = \sum_{j=1}^{F} \left(\sum_{i=1}^{N} |\mathcal{A}_{ij} - \mathcal{B}_{ij}|^2 \cdot \mu_i\right),$$

the claim is established. $\square$

## H  PROOF OF THEOREM 4.1

Here we prove Theorem 4.1, which we restate here for convenience:

**Theorem H.1.** Suppose for filter banks $\{\Psi_i^{\text{fwd/fwd}}\}_{I^{\text{fwd/fwd}}}$ that the matrices $\Psi_i^{\text{fwd}}(T)$, $\Psi_i^{\text{bwd}}(T^*)$ contain only real entries. Then any HoloNet with layer-widths $\{F_\ell\}$ with complex weights & biases and a non linearity that acts on complex numbers componentwise as $\rho(a + ib) = \widetilde{\rho}(a) + i\widetilde{\rho}(a)$ can be exactly represented by a HoloNet of widths $\{2 \cdot F_\ell\}$ utilizing $\widetilde{\rho}$ and employing only real weights & biases.

*Proof.* It suffices to prove, that in this setting the update rule

$$X^\ell = \rho\left(\alpha \sum_{i \in I} \Psi_i^{\text{fwd}}(T) \cdot X^{\ell-1} \cdot W_i^{\text{fwd},\ell} + (1 - \alpha) \sum_{i \in I} \Psi_i^{\text{bwd}}(T^*) \cdot X^{\ell-1} \cdot W_i^{\text{bwd},\ell} + B^\ell\right).$$

can be replaced by purely real weights and biases. For simplicity in notation, let us assume $\alpha = 1$; the general case follows analogously but with more cluttered notation.

Let us write $X = X_{real} + iX_{imag}$, $W = W_{real} + iW_{imag}$, $B = B_{real} + iB_{imag}$. Then we have with $\{\Psi_i^{\text{fwd}}(T)\}_{i \in I}$ purely real, that

$$X^\ell = \rho\left(\sum_{i \in I} \Psi_i^{\text{fwd}}(T) \cdot X^{\ell-1} \cdot W_i^{\text{fwd},\ell} + B^\ell\right)$$

$$= \rho\left(\sum_{i \in I} \Psi_i^{\text{fwd}}(T) \cdot (X_{real}^{\ell-1} W_{real,i}^{\text{fwd},\ell} - X_{imag}^{\ell-1} W_{imag,i}^{\text{fwd},\ell}) + B_{real}^\ell\right.$$

$$\left. +i\left[\sum_{i \in I} \Psi_i^{\text{fwd}}(T) \cdot (X_{real}^{\ell-1} W_{imag,i}^{\text{fwd},\ell} + X_{imag}^{\ell-1} W_{real,i}^{\text{fwd},\ell}) + B_{imag}^\ell\right]\right)$$

$$= \widetilde{\rho}\left(\sum_{i \in I} \Psi_i^{\text{fwd}}(T) \cdot (X_{real}^{\ell-1} W_{real,i}^{\text{fwd},\ell} - X_{imag}^{\ell-1} W_{imag,i}^{\text{fwd},\ell}) + B_{real}^\ell\right)$$

$$+ i\widetilde{\rho}\left(\sum_{i \in I} \Psi_i^{\text{fwd}}(T) \cdot (X_{real}^{\ell-1} W_{imag,i}^{\text{fwd},\ell} + X_{imag}^{\ell-1} W_{real,i}^{\text{fwd},\ell}) + B_{imag}^\ell\right)$$

The result then immediately follows after using the canonical isomorphism between $\mathbb{C}^d$ and $\mathbb{R}^{2d}$ as

$$X^\ell \cong \begin{pmatrix} X_{real}^\ell \\ X_{imag}^\ell \end{pmatrix} = \widetilde{\rho}\left[\begin{pmatrix} \sum_{i \in I} \Psi_i^{\text{fwd}}(T) \cdot (X_{real}^{\ell-1} W_{real,i}^{\text{fwd},\ell} - X_{imag}^{\ell-1} W_{imag,i}^{\text{fwd},\ell}) + B_{real}^\ell \\ \sum_{i \in I} \Psi_i^{\text{fwd}}(T) \cdot (X_{real}^{\ell-1} W_{imag,i}^{\text{fwd},\ell} + X_{imag}^{\ell-1} W_{real,i}^{\text{fwd},\ell}) + B_{imag}^\ell \end{pmatrix}\right]. \quad (17)$$

The above layer update

$$\begin{pmatrix} X^{\ell-1}_{real} \\ X^{\ell-1}_{imag} \end{pmatrix} \overset{(17)}{\longmapsto} \begin{pmatrix} X^{\ell}_{real} \\ X^{\ell}_{imag} \end{pmatrix}$$

can then clearly be realised by a real network as described in Theorem 4.1.

$\square$

## I  ADDITIONAL DETAILS ON EXPERIMENTS:

### I.1  FABERNET: NODE CLASSIFICATION

**Datasets:**  We evaluate on the task of node classification on several directed benchmark datasets with high homophily: Chameleon & Squirrel (Pei et al., 2020), Arxiv-Year (Hu et al., 2020b), Snap-Patents (Lim et al., 2021) and Roman-Empire (Platonov et al., 2023). These datasets are highly heterophilic (edge homophily smaller than 0.25).

Table 5: Node Classification Datasets: Statistics

| DATASET | # NODES | # EDGES | # FEAT. | # C | UNID. EDGES | EDGE HOM. |
|---|---|---|---|---|---|---|
| CHAMELEON | 2,277 | 36,101 | 2,325 | 5 | 85.01% | 0.235 |
| SQUIRREL | 5,201 | 217,073 | 2,089 | 5 | 90.60% | 0.223 |
| ARXIV-YEAR | 169,343 | 1,166,243 | 128 | 40 | 99.27% | 0.221 |
| SNAP-PATENTS | 2,923,922 | 13,975,791 | 269 | 5 | 99.98% | 0.218 |
| ROMAN-EMPIRE | 22,662 | 44,363 | 300 | 18 | 65.24% | 0.050 |

**Experimental Setup**  All experiments are conducted on a machine with NVIDIA A4000 GPU with 16GB of memory, safe for experiments on snap-patents which have been performed on a machine with one NVIDIA Quadro RTX 8000 with 48GB of memory. We closely follow the experimental setup of (Rossi et al., 2023). In all experiments, we use the Adam optimizer and train the model for 10000 epochs, using early stopping on the validation accuracy with a patience of 200 for all datasets apart from Chameleon and Squirrel, for which we use a patience of 400. For OGBN-Arxiv we use the fixed split provided by OGB (Hu et al., 2020b), for Chameleon and Squirrel we use the fixed GEOM-GCN splits (Pei et al., 2020), for Arxiv-Year and Snap-Patents we use the splits provided in Lim et al. (2021), while for Roman-Empire we use the splits from Platonov et al. (2023).

**Baselines Results:**  Results for MLP, GCN, $H_2$GCN, GPR-GNN and LINKX were taken from Lim et al.. Results for Gradient Gating are taken from their paper (Rusch et al., 2023). Results for GloGNN are taken from their paper (Li et al., 2022). Results on Roman-Empire are taken from Platonov et al. (2023) for GCN, $H_2$GCN, GPR-GNN, FSGNN and GloGNN and from Rossi et al. (2023) for MLP, LINKX, ACM-GCN and Gradient Gating. Results for FSGNN are taken from Maurya et al. (2021) for Actor, Squirrel and Chameleon, and from Rossi et al. (2023) for results on Arxiv-year and Snap-Patents. Results for DiGCN, MagNet are taken from Rossi et al. (2023). Results for DirGNN were obtained via a re-implementation; using the official codebase and hyperparamters specified in (Rossi et al., 2023). Note that – as detailed in Rossi et al. (2023) – the reported results for DirGNN correspond to a best-of-three report over directed version of GCN (Kipf & Welling, 2017), GAT (Velickovic et al., 2018) and Sage (Hamilton et al., 2017).

**Design Choices:**  Some design choices made in Section (4) warrant an additional discussion:

- **Separation of forward and backward filters**
- **Choice of zero-diagonal for** $T$
- **Choice of normalization** in $T$ as $T = (D^{in})^{-\frac{1}{4}} \cdot W \cdot (D^{out})^{-\frac{1}{4}}$ as opposed to the more traditional square-root normalization in $T = (D^{in})^{-\frac{1}{2}} \cdot W \cdot (D^{out})^{-\frac{1}{2}}$.
- **Choice of non-linearity**: $|\cdot|$ vs ReLu.

We here provide the corresponding discussions:

In principle, the need to consider both **forward- and backward-filters** arises from the nature of directed graphs: If – say– only forward filters would be implemented, information would only be allowed to flow *along* the directed edges of a given graphs. In a citation network (with the characteristic operators $T$ chosen e.g. as the adjacency matrix or a graph Laplacian) for example, this would only endow a given node representing a specific paper with information about papers that the given paper cites. The (equally important) information about papers that cite the given one never reaches the node of our fixed paper. This can be remedied by not only considering filters based on $T$, but also its transpose $T^*$: One can consider $T^*$ as describing the original graph, with all edge directions reversed, so that $T^*$ facilitates information flow in exactly the opposite direction when compared with $T$. In principle, one could then be tempted try to implement filters as functions $f(x, y)$ in two complex variables ($x$ and $y$) and make the replacements $z \mapsto T$ and $y \mapsto T^*$, thus implementing a joint filter $f(T, T^*)$ for both directions simultaneously. However, such a procedure is unfortunately mathematically ill-defined Kato (1976): Consider for example the function $f(z, y) = zy = yz$. It is then not clear if we should set $f(T, T^*) = T \cdot T^*$ or $f(T, T^*) = T^* \cdot T$. This would not be a problem if $T^* \cdot T = T \cdot T^*$ and one may indeed prove that in this setting, one might consistently define the matrix $f(T, T^*)$; the relevant tool here is the "Functional calculus for commuting operators". In general however, we have $T^* \cdot T \neq T \cdot T^*$ if $T$ describes a directed graph. Hence we can not construct such a joint spectral filter $f(T, T^*)$ and instead consider learnable spectral filters of the form $f(T) + g(T^*)$. In principle, one may of course also consider other algebraic combination of the two directions of information flow (such as e.g. $f(T) \cdot g(T^*)$ and $g(T^*) \cdot f(T)$). We however leave this for future work.

To avoid comparing the features of surrounding nodes with the feature vector of a given node when applying the characteristic operator $T$ (or powers $T^k$ of $T$) to the feature matrix $X$ as $T \cdot X$, **we choose $T$ to have only the entry zero on the diagonal**. This is done, as FaberNet is designed for node-classification on *heterophilic* graphs. Thus, when updating the feature vector of a given node, the previous feature vector of this node is discarded, and the new feature vector is solely made up of information about surrounding nodes (i.e. information about the neighbourhood(structure) of the original node). This avoids a mixing of self-features and neighbourhood features, which is desireable as Maurya et al. (2021) pointed out theoretically, and we also observed empirically in our experiments. Hence we choose $T$ as (some form of) the adjacency matrix, as it has a zero-diagonal.

Different from the **normalization** of $T$ that we have chosen, a symmetric normalization such as $T = (D^{in})^{-\frac{1}{2}} \cdot W \cdot (D^{out})^{-\frac{1}{2}}$. might arguably be considered more traditional. Historically, this can be traced back to the utilization of the symmetrically normalized Laplacian as providing the Fourier-atoms (i.e. the Laplacian eigenvectors) of the graph Fourier Transform that underlies the original undirected spectral models such as Bruna et al. (2014); Defferrard et al. (2016); Kipf & Welling (2017). A main point of the present paper however, is transcending this graph Fourier transform, as the traditional approach is limited to undirected graphs. Our approach instead does not need to be based on any underlying Laplacian on the graph: Indeed, the holomorphic functional calculus may be applied to arbitrary operators. As such, a traditional holding-on-to a normalization stemming from a traditional use of a Laplacian-based Graph-Fourier transforms is no longer necessary. As we observed experimentally, such a traditional normalization is also (slightly) disadvantageous, as far as performance is concerned.

Arguably, a majority of the machine learning literature in general uses some form of (leaky-) ReLU activations. In the graph setting, an exception is constituted by graph-scattering-networks (c.f. e.g. Perlmutter et al. (2019); Koke & Kutyniok (2022)): Corresponding scattering architectures are based on the absolute-value-nonlinearity. For us, the **choice to use either ReLu or the absolute value** arises as our networks are potentially complex. In the complex setting there is no clear consensus which activation function performs best (c.f. e.g. Lee et al. (2022)). Hence we consider two possible choices in our architecture, whenever we enable complex parameters (c.f. also Table 7 below).

**Hyperparameters:** Following the setup of Rossi et al. (2023), our search space for generic hyper-parameters is given by varying the learning rate $lr \in \{0.01, 0.005, 0.001, , 0.0005\}$, the hidden dimension over $F \in \{32, 64, 128, 256, 512\}$, ne number of layers over $L \in \{2, 3, 4, 5, 6\}$, jumping knowledge connections over $jk \in \{max, cat, none\}$ layer-wise normalization in $norm \in \{True, False\}$, patience as $patience \in \{200, 400\}$ and dropout as $p \in \{0, 0.2, 0.4, 0.6, 0.8, 1\}$.

Table 6: Final Generic Hyperparameters

| DATASET | $lr$ | L | PATIENCE | F | NORM | P | JK |
|---|---|---|---|---|---|---|---|
| CHAMELEON | 0.005 | 5 | 400 | 128 | TRUE | 0 | CAT |
| SQUIRREL | 0.01 | 4 | 400 | 128 | TRUE | 0 | MAX |
| ARXIV-YEAR | 0.005 | 6 | 200 | 256 | FALSE | 0 | CAT |
| SNAP-PATENTS | 0.01 | 5 | 200 | 32 | TRUE | 0 | MAX |
| ROMAN-EMPIRE | 0.01 | 5 | 200 | 256 | FALSE | 0.2 | CAT |

In practice we take the parameters of Table 6 as frozen and given by Rossi et al. (2023). We then optimize over the custom hyperparameters pertaining to our method. To this end, we vary the maximal order of our Faber polynomials $\{\Psi_i\}_{i=0,1}^K$ as $K \in \{1, 2, 3, 4, 5\}$. Note that we also discount higher order terms with a regularization $\sim 1/2^i$, as this improved results experimentally. We thus have $\Psi_k = \lambda^k/2^k$. The type of weights & bisases is varied over $parameters \in \{\mathbb{R}, \mathbb{C}\}$. The non-linearity is varied over $\{|\cdot|_{\mathbb{C}}, \text{ReLu}\}$, with $|a + ib|_{\mathbb{C}} = |a| + i|b|$. The parameter $\alpha$ is varied as $\alpha \in \{0, 0.5, 1\}$ as in Rossi et al. (2023). The zero-order Faber polynomial $\Psi_0(\lambda) = 1$ is either included or discarded, as discussed in Section 4 and weight decay parameters for real- and imaginary weights are varied over $\lambda_{\text{real}}, \lambda_{\text{imag}} \in \{0, 0.1, 1\}$. Final selected hyperparameters are listed in Table 7.

Table 7: Final Custom Hyperparameters

| DATASET | $K$ | PARAMETERS | NON.-LIN. | $\alpha$ | $\Psi_0$ | $\lambda_{\text{REAL}}$ | $\lambda_{\text{IMAG}}$ |
|---|---|---|---|---|---|---|---|
| CHAMELEON | 4 | $\mathbb{C}$ | $|\cdot|_{\mathbb{C}}$ | 0 | No | 1 | 0 |
| SQUIRREL | 5 | $\mathbb{C}$ | $|\cdot|_{\mathbb{C}}$ | 0 | No | 0.1 | 0.1 |
| ARXIV-YEAR | 1 | $\mathbb{R}$ | ReLU | 0.5 | No | 0.1 | N.A. |
| SNAP-PATENTS | 2 | $\mathbb{R}$ | ReLU | 0.5 | No | 0.1 | N.A. |
| ROMAN-EMPIRE | 1 | $\mathbb{R}$ | ReLU | 0.5 | Yes | 0.1 | N.A. |

## I.2 DIR-RESOLVNET: DIGRAPH REGRESSION AND SCALE INSENSITIVITY

**Dataset:** The dataset we consider is the **QM7** dataset, introduced in Blum & Reymond (2009); Rupp et al. (2012). This dataset contains descriptions of 7165 organic molecules, each with up to seven heavy atoms, with all non-hydrogen atoms being considered heavy. In the dataset, a given molecule is represented by its Coulomb matrix $W$, whose off-diagonal elements

$$W_{ij} = \frac{Z_i Z_j}{|\vec{x}_i - \vec{x}_j|} \tag{18}$$

correspond to the Coulomb-repulsion between atoms $i$ and $j$. We discard diagonal entries of Coulomb matrices; which would encode a polynomial fit of atomic energies to nuclear charge (Rupp et al., 2012).

To each molecule an atomization energy - calculated via density functional theory - is associated. The objective is to predict this quantity. The performance metric is mean absolute error. Numerically, atomization energies are negative numbers in the range $-600$ to $-2200$. The associated unit is [*kcal/mol*].

For each atom in any given molecular graph, the individual Cartesian coordinates $\vec{x}_i$ and the atomic charge $Z_i$ are also accessible individually.

In order to induce directedness, we modify the Coulomb weights (18): Weights *only* from heavy atoms to *atoms outside this heavy atom's respective immediate hydrogen cloud* are modified as

$$W_{ij} := \frac{Z_i^{\text{outside}} \cdot (Z_j^{\text{heavy}} - 1)}{|\vec{x}_i - \vec{x}_j|}. \tag{19}$$

The immediate hydrogen cloud of a given heavy atom, we take to encompass precisely those hydrogen atoms for which this heavy atom is the closest among all other heavy atoms.

This specific choice (19) is made in preparation for the scale insensitivity experiments: The theory developed in Section 3.3.2 applies to those graphs, where strongly connected subgraphs contain only nodes for which the in-degree equals the out-degree (where only strong weights are considered when calculating the respective degrees). The choice (19) facilitates this, as hydrogen-hydrogen weights and weights between hydrogen and respective closest heavy atom remain symmetric.

**Experimental Setup:** We shuffle the dataset and randomly select 1500 molecules for testing. We then train on the remaining graphs. We run experiments for 5 different random random seeds and report mean and standard deviation.

All considered convolutional layers (i.e. for Dir-ResolvNet and baselines) are incorporated into a two layer deep and fully connected graph convolutional architecture. In each hidden layer, we set the width (i.e. the hidden feature dimension) to

$$F_1 = F_2 = 64.$$

For all baselines, the standard mean-aggregation scheme is employed after the graph-convolutional layers to generate graph level features. Finally, predictions are generated via an MLP.

For Dir-ResolvNet, we take $\alpha = 1$, use real weights and biases and set $z = -1$. These choices are made for simplicity. Resolvents are thus given as

$$R_{-1}(\Delta) = (\Delta + Id)^{-1}.$$

As aggregation for our model, we employ the graph level feature aggregation scheme $\Omega$ introduced before Theorem 4.2 in Section 4. Node weights set to atomic charges of individual atoms. Predictions are then generated via a final MLP with the same specifications as the one used for baselines.

**Scale Insensitivity** We then modify (all) molecular graphs in QM7 by deflecting hydrogen atoms (H) out of their equilibrium positions towards the respective nearest heavy atom. This is possible since the QM7 dataset also contains the Cartesian coordinates of individual atoms.

This introduces a two-scale setting precisely as discussed in section 3.3.2: Edge weights between heavy atoms remain the same, while Coulomb repulsions between H-atoms and respective nearest heavy atom increasingly diverge; as is evident from (18).

Given an original molecular graph $G$ with node weights $\mu_i = Z_i$, the corresponding limit graph $\underline{G}$ corresponds to a coarse grained description, where heavy atoms and surrounding H-atoms are aggregated into single super-nodes in the sense of Section 3.3.2.

Mathematically, $\underline{G}$ is obtained by removing all nodes corresponding to H-atoms from $G$, while adding the corresponding charges $Z_H = 1$ to the node-weights of the respective nearest heavy atom. Charges in (19) are modified similarly to generate the weight matrix $\underline{W}$.

On original molecular graphs, atomic charges are provided via one-hot encodings. For the graph of methane – consisting of one carbon atom with charge $Z_C = 6$ and four hydrogen atoms of charges $Z_H = 1$ – the corresponding node-feature-matrix is e.g. given as

$$X = \begin{pmatrix} 0 & 0 & \cdots & 0 & 1 & 0\cdots \\ 1 & 0 & \cdots & 0 & 0 & 0\cdots \\ 1 & 0 & \cdots & 0 & 0 & 0\cdots \\ 1 & 0 & \cdots & 0 & 0 & 0\cdots \\ 1 & 0 & \cdots & 0 & 0 & 0\cdots \end{pmatrix}$$

with the non-zero entry in the first row being in the 6$^{\text{th}}$ column, in order to encode the charge $Z_C = 6$ for carbon.

The feature vector of an aggregated node represents charges of the heavy atom and its neighbouring H-atoms jointly.

As discussed in Section 3.3.2, node feature matrices are translated as $\underline{X} = J^{\downarrow}X$. Applying $J^{\downarrow}$ to one-hot encoded atomic charges yields (normalized) bag-of-word embeddings on $\underline{G}$: Individual entries of feature vectors encode how much of the total charge of the super-node is contributed by individual atom-types. In the example of methane, the limit graph $\underline{G}$ consists of a single node with node-weight

$$\mu = 6 + 1 + 1 + 1 + 1 = 10.$$

The feature matrix

$$\underline{X} = J^{\downarrow} X$$

is a single row-vector given as

$$\underline{X} = \left( \frac{4}{10}, 0, \cdots, 0, \frac{6}{10}, 0, \cdots \right).$$

### I.3 EFFICIENCY ANALYSIS:

Here we compare the efficiency of our model with the directed baselines of Section 5.

**Complexity:** For definiteness, we will assume the DirGCN configuration of the DirGNN baseline (other configurations would yield higher complexity). For a given graph $G$, let $N$ denote its number of nodes and let $E$ denote its number of edges. Assume we have input features of dimension $C$ and a $F$ hidden dimensions in our architectures.

DiGCN: In its full form, DiGCN has a $\mathcal{O}(N^2)$ complexity, which is reduced to a $\mathcal{O}(EFC)$ complexity in its approximate propagation scheme, which is used in practice.

DirGNN: Similarly, DirGNN possesses a $\mathcal{O}(EFC)$. In both cases this stems from the sparse-dense matrix multiplications that facilitate the forward function.

MagNet: MagNet is similarly implemented via sparse-dense matrix multiplications; resulting in an $\mathcal{O}(EFC)$-complexity.

FaberNet: Since FaberNet too implements its forward using sparse-dense matrix multiplications, its filtering operation also has $\mathcal{O}(EFC)$ complexity.

Dir-ResolvNet: The filtering operation in Dir-ResolvNet is implemented as dense-dense matrix multiplication, and as such has complexity $\mathcal{O}(N^2)$ like the full (not approximated) forward operation of DiGCN would have.

**Number of trainable parameters:** Compared to DirGNN (in its DirGCN Setting) and assuming the same network width and depth, our FaberNet has $K$ times the number of learnable parameters in the real setting. In the complex setting, our method has $2 \cdot K$ as many *real* parameters as DirGNN. Here $K$ refers to the highest utilized order of Faber polynomials $\{\Psi_k\}_k$. In our experiments, the maximal attained value of $K$ was $K = 5$ on the "Squirrel" dataset (c.f. Table 7). On this dataset, complex weights & biases performed best, so that our method has 6.3 M trainable parameters with the configuration specified in Table 6.

## J  PSEUDOCODE

In this section we provide a pseudocode description of the forward function implemented within a standard Holonet. Using the notation of Section 4, we first recall the corresponding update rule:

$$X^\ell = \rho \left( \alpha \sum_{i \in I^{\text{fwd}}} \Psi_i^{\text{fwd}}(T) \cdot X^{\ell-1} \cdot W_i^{\text{fwd},\ell} + (1 - \alpha) \sum_{i \in I^{\text{bwd}}} \Psi_i^{\text{bwd}}(T^*) \cdot X^{\ell-1} \cdot W_i^{\text{bwd},\ell} + B^\ell \right).$$
(20)

In practice, a full forward pass within a standard HoloNet is then implemented as described in Algorithm 1 below:

---

**Algorithm 1:** The forward function of HoloNets

---

**Data:**  Basis functions $\Psi_i^{\text{fwd}}(T), \Psi_i^{\text{bwd}}(T^*)$ evaluated at $T, T^*$ (precomputed; for
$1 \leqslant i \leqslant M^{\text{fwd/bwd}}$); Network depth $L \in \mathbb{N}$; Feature dimensions $F_\ell$ (for $0 \leqslant \ell \leqslant L$);
Weighting parameter $\alpha \in [0, 1]$; Biases $B_i^{\text{fwd/bwd},\ell}$ (for $1 \leqslant i \leqslant M^{\text{fwd/bwd}}$ and $1 \leqslant \ell \leqslant L$);
Weight matrices $W_i^{\text{fwd/bwd},\ell}$ of dimension $F_\ell \times F_{\ell+1}$ (for $1 \leqslant i \leqslant M^{\text{fwd/bwd}}$ and
$1 \leqslant \ell \leqslant L$)
**Input:** Node-feature matrix $X$
**Output:** Final-layer-features $H^L$
$H^0 \leftarrow X$;
**for** $1 \leqslant \ell \leqslant L$ **do**
    $H^\ell \leftarrow \alpha \cdot Linear(matmul(\Psi_1^{\text{fwd}}(T), H^{\ell-1}), W_1^{\text{fwd},\ell}, B_1^{\text{fwd},\ell})$;
    $H^\ell \leftarrow H^\ell + (1 - \alpha) \cdot Linear(matmul(\Psi_1^{\text{bwd}}(T^*), H^{\ell-1}), W_1^{\text{fwd},\ell}, B_1^{\text{fwd},\ell})$;
    **if** $M^{fwd} \geqslant 2$ **then**
        **for** $2 \leqslant i \leqslant M^{fwd}$ **do**
            $H^\ell \leftarrow H^\ell + \alpha \cdot Linear(matmul(\Psi_i^{\text{fwd}}(T), H^{\ell-1}), W_i^{\text{fwd},\ell}, B_i^{\text{fwd},\ell})$;
        **end**
    **end**
    **if** $M^{bwd} \geqslant 2$ **then**
        **for** $2 \leqslant i \leqslant M^{bwd}$ **do**
            $H^\ell \leftarrow H^\ell + (1 - \alpha) \cdot Linear(matmul(\Psi_i^{\text{bwd}}(T^*), H^{\ell-1}), W_1^{\text{bwd},\ell}, B_i^{\text{bwd},\ell})$;
        **end**
    **end**
    $H^\ell \leftarrow \rho \left( H^\ell \right)$
**end**
**return** $H^L$

---

Note that the bias term $B^\ell$ in (20) is related to the individual bias terms in Algorithm 1 as

$$B^\ell = \sum_{i=1}^{M^{\text{fwd}}} B_i^{\text{fwd}} + \sum_{i=1}^{M^{\text{bwd}}} B_i^{\text{bwd}},$$

where biases are broadcasted to all $N$ nodes to match the dimension of $H^\ell$.

