# OpenReview forum: "HoloNets: Spectral Convolutions do extend to Directed Graphs"
_ICLR.cc/2024/Conference — ICLR 2024 poster_

### Official Review · Reviewer_XGUM · 2023-10-27

**Soundness:** 3 good
**Presentation:** 2 fair
**Contribution:** 3 good
**Rating:** 6
**Confidence:** 3

**Summary:**

This paper introduces spectral convolutions to directed graphs without relying on the graph Fourier transform. The paper provides a frequency-response interpretation of the proposed filters with a theoretical analysis. Experimental results seem promising.

**Strengths:**

1) The usage of holomorphic functional calculus to introduce directed graph filters is novel.
2) Directed graphs are important to consider.
3) Theoretical analysis seems sound.
4) The proposed method seems effective and expressive.

**Weaknesses:**

1) The formulation seems to be restricted to one-dimensional node weights for the nodes, but it is not clear what modifications should be made if we are given multiple-dimensional node features.
2) The underlying reason for using holomorphic functional calculus has not been clearly demonstrated.
3) Some typos exist. For example, in the "Contributions" part at the end of Sec. 1, there is one redundant "the" in line 6; in the second paragraph of Sec. 3.3.1, there is one more "near" in line 5; there should be an 'as' after 'referred to' right after equation (5) on page 17; the first 'eigenvalue' should indeed be 'eigenvector' in Sec. C on page 17; 'interpreting' should be 'interpreted' right after the mention of Appendix B on page 18.

**Questions:**

1) The formulation seems to be restricted to one-dimensional node weights for the nodes. What modifications should be made if we are given multiple-dimensional node features?
2) What is the computational complexity of HoloNets compared to DiGCN, MagNet, and Dir-GNN?

---

> ### Author Response · Authors · 2023-11-18
> **Provided additional Details on the Necessity of the Holomorphic Functional Calculus and added a Discussion of Computational Complexity**
>
> We sincerely thank the reviewer for the careful consideration of our paper! We were especially happy to read  the sentiments that
> > the usage of holomorphic functional calculus to introduce directed graph filters is novel,
>
> that our
> >  theoretical analysis seems sound
>
>  and our
> > proposed method seems effective and expressive.
>
> Let us adress the raised points individually below:
>
> * The formulation seems to be restricted to one-dimensional node weights for the nodes, but it is not clear what modifications should be made if we are given multiple-dimensional node features.
>
> Channel mixing in spectral graph networks is implemented in analogy to the Euclidean case: A Euclidean convolutional filter applied to an image acts only on a single (r, g or b)-channel. Channel mixing is then facilitated by connecting all channels in two consecutive layers via convolutional filters. All incoming information at a given channel is then summed up. A convenient visual representation may e.g. be found in Figure 2 of (Koke (2023)) .
>
> Our method follows this paradigm. This might for example be explicitly seen in the layer update rule in Section 4: As described there, the  weight matrices $W_k^{\text{fwd/bwd},\ell+1}$ are of dimension $F_{\ell}\times F_{\ell+1}$, with $F_{\ell}, F_{\ell+1}$ the feature dimensions of the respective layers.
> These matrices implement the contact between different channels.
>
> * "The underlying reason for using holomorphic functional calculus has not been clearly demonstrated."
>
> The guiding principle behind modern spectral convolutional methods is that of designing filters of the form $g(T)$, with $T$ a characteristic operator and $g$ a learnable (scalar) function (Defferrard et. al (2016)). In the undirected setting, we had $T = T^*$, which – as we detail in Sections 1 through 3 – allows to unambiguously evaluate the scalar function $g$ on the matrix $T$ by essentially evaluating $g$ on each eigenvalue of $T$ independently.
>
> As we detail in Sections 2 and 3, this is no longer possible in the directed setting; instead the only way to consistently define the matrix $g(T)$ is to make use of the holomorphic functional calculus.
>
> Following this concern raised by the reviewer, we have modified the corresponding discussion of this matter in Section 3.1,  to reflect even more strongly that $g(T)$ may only be consistently defined if $g$ is assumed to be holomorphic.
>
> If the reviewer would however like us to incorporate additional specific comments on this matter, we would be more than happy to do so.
>
>
>
> * "Some typos exist. "
>
> We are very grateful to the reviewer spotting these typos! We have dilligenly fixed them all in the updated version of our manuscript.
>
> * "What is the computational complexity of HoloNets compared to DiGCN, MagNet, and Dir-GNN?"
>
> For definiteness, we will assume the DirGCN configuration of the DirGNN baseline (other configurations would yield higher complexity). For a given graph $G$, let $N$ denote its number of nodes and let $E$ denote its number of edges. Assume  we have input features of dimension $C$ and  a $F$ hidden dimensions in our architectures.
>
> DiGCN: In its full form, DiGCN has a $\mathcal{O}(N^2)$ complexity, which is reduced to a  $\mathcal{O}(EFC)$ complexity in its approximate propagation scheme, which is used in practice.
>
> DirGNN: Similarly, DirGNN possesses a $\mathcal{O}(EFC)$. In both cases this stems from the sparse-dense matrix multiplications that facilitate the forward function.
>
> MagNet: MagNet is similarly implemented via sparse-dense matrix multiplications; resulting in an $\mathcal{O}(EFC)$-complexity.
>
> FaberNet: Since FaberNet too implements its forward using sparse-dense matrix multiplications, its filtering operation also has $\mathcal{O}(EFC)$ complexity.
>
> Dir-ResolvNet: The filtering operation in Dir-ResolvNet is implemented as dense-dense matrix multiplication, and as such has complexity $\mathcal{O}(N^2)$ like the full (not approximated) forward operation of DiGCN would have.
>
>
> Following this question, we have also included this discussion in a new section (Section I.3: Efficiency Analysis) in our updated manuscript.

---

> > ### Comment · Reviewer_XGUM · 2023-11-20
> > **Thank you**
> >
> > Thank you for your response. I am happy to keep my current score to support the paper's acceptance.

---

### Official Review · Reviewer_T8Vv · 2023-10-29

**Soundness:** 3 good
**Presentation:** 3 good
**Contribution:** 4 excellent
**Rating:** 8
**Confidence:** 2

**Summary:**

This paper targets the limitation of conventional Graph Neural Networks (GNNs) that struggle with handling directed graphs effectively. The objective is to enhance spectral convolution networks to cater to directed graphs, thereby eliminating reliance on the Graph Fourier Transform. The proposed method's effectiveness is confirmed through experiments conducted on real-world datasets.

**Strengths:**

1. The paper is novel and has theoretical depth.
2. The proposed method is technically sound.
3. The paper is well-written and easy to follow.
4. The proposed method achieves SOTA performance in several real-world directed heterophilic datasets.

**Weaknesses:**

I am not an expert in spectral GNNs thus I could only point out a limited number of issues in the experiments:
1. It would be good to examine the efficiency of the proposed method, especially when compared with DiGCN and DirGNN. Moreover, a discussion on the number of learnable parameters would be beneficial.
2. The impact of the number of layers on the model's performance is worth exploring. Do we have the same problem of over-smoothing in directed graphs when the GNN goes deeper?
3. Did HoloNet outperform the baselines in graph classification tasks? More evaluation tasks could be investigated.


Overall, I think it's a good paper ready for publication at NeurIPS.

**Questions:**

Please reply to questions in weaknesses.

---

> ### Author Response · Authors · 2023-11-18
> **Added a discussion of efficiency as well as number of learnable parameters**
>
> We would like to sincerely thank the reviewer for the careful evaluation  of our paper and appreciation of our results. We were especially  happy to read that
> > the paper is well written and easy to follow,
>
> >“novel and has theoretical depth
>  and is
>
> > ready for publication at NeurIPS.
>
> Let us address the raised points individually:
>
> * It would be good to examine the efficiency of the proposed method, especially when compared with DiGCN and DirGNN.
>
> We are very happy to do so:
>
> For definiteness, we will assume the DirGCN configuration of the DirGNN baseline (other configurations would yield higher baseline-complexity). For a given graph $G$, let $N$ denote its number of nodes and let $E$ denote its number of edges. Assume  we have input features of dimension $C$ and   $F$ hidden dimensions in each layer of our architectures.
>
> DiGCN: In its full form, DiGCN has a $\mathcal{O}(N^2)$ complexity, which is reduced to a  $\mathcal{O}(EFC)$ complexity in its approximate propagation scheme, which is used in practice.
>
> DirGNN: Similarly, DirGNN possesses a $\mathcal{O}(EFC)$ complexity. In both cases this stems from the sparse-dense matrix multiplications that facilitate the forward process.
>
> FaberNet: Since FaberNet too implements its forward using sparse-dense matrix multiplications, its filtering operation also has $\mathcal{O}(EFC)$ complexity.
>
> Dir-ResolvNet: The filtering operation in Dir-ResolvNet is implemented as dense-dense matrix multiplication, and as such has complexity $\mathcal{O}(N^2)$ like the full (not approximated) forward operation of DiGCN would have.
>
> Following this point raised by the reviewer, we have now included this discussion into a new subsection titled “Efficiency Analysis” in "Appendix I: Additional Details  on Experiments” in our revised manuscript.
> * "Moreover, a discussion on the number of learnable parameters would be beneficial."
>
> Compared to DirGNN (in its DirGCN Setting) and assuming the same network width and depth, our FaberNet has $K$ times the number of learnable parameters in the real setting. In the complex setting, our method has $2\cdot K$ as many _real_ parameters as DirGNN.  Here $K$ refers to the highest utilized order of Faber polynomials $\Psi_k$. In our experiments, the maximal attained value of $K$ was $K = 5$ on  the “Squirrel” dataset. On this dataset, complex weights and biases performed best, so that our method has 6.3 M  trainable parameters in the corresponding utilized hyperparameter-configuration for Squirrel (detailed in Appendix I).
>
> We have included this discussion above into the newly created subsection titled “Efficiency Analysis” in "Appendix I: Additional Details  on Experiments” in our revised manuscript.

---

> > ### Author Response · Authors · 2023-11-18
> > **Discussing additional points and potential Future Work**
> >
> > * More evaluation tasks could be investigated.
> >
> > In principle, we agree that more evaluation tasks could potentially be investigated. However, in an effort to not overload the (in our opinion) already quite information-rich paper, we limited ourselves to one task at the node-level and one task at the graph-level.
> >
> > Our aim in doing so is to showcase the potential, stability and expressivity  of directed spectral convolutional filters for both theseexperimental settings. We hope that this will result in a swift uptake of our HoloNet framework by the community and a further exploration of this research direction, with followup works on architectures e.g. adapted to additional tasks beyond the ones we already considered within our work.
> >
> >
> > * The impact of the number of layers on the model's performance is worth exploring. Do we have the same problem of over-smoothing in directed graphs when the GNN goes deeper?
> >
> >
> > We agree that this is an interesting future research direction.
> >
> > In our experiments, we already go up to layer-depth $6$, which is already deeper than the optimal depth for e.g. GCN. We will be glad to include a detailed ablation study on the number of layers in a camera ready version of our submission.
> >
> > Here, we provide our theoretical opinion on this matter:
> >
> > In the undirected setting, the degradation of feature-quality with layer depth is related to Laplacian smoothing. This smoothing operation makes node representations more and more similar as a signal is propagated through the layers (c.f. e.g. [1]).
> >
> > Mathematically, Laplacian smoothing can be understood as mapping an original feature matrix $X$ to its smoothed version $X \mapsto e^{-\Delta \cdot t}\cdot X$. Here $\Delta$ denotes a suitable graph Laplacian and the continuous time $t$ is replaced by the layer-depth $\ell$ when discussing the low-pass behaviour of GNNs. Smoothing now occurs, bevause the function $\lambda \mapsto e^{-\lambda\cdot t}$ suppresses all non-zero eigenvalues $\lambda$; especially as $t$ (or equivalently the layer-depth) is increased.
> >
> > For large times ($t \gg 1$) or respectively deep architectures, the only information that survives is that corresponding to the $\lambda = 0$ eigenvalue. Since the corresponding eigenvector is simply given (up to normalization) as $v_0^T = (1,…,1)^T$ (assuming that the graph is connected), this means that only a graph-average  $v_0^T\cdot X$ of all the available information $X$ survives.
> >
> > In the directed setting, it is possible that the undirected version of a directed graph is connected, while the original directed graph contains more than one _reach_ (c.f. the Discussion of reaches in Section 2 of our submission). In this case, there is no longer a single eigenvector associated to the eigenvalue $0$ of the (directed) graph Laplacian. Instead, the multiplicity of the eigenvalue $0$ is given by the number of distinct reaches (c.f. e.g. [2] ). Thus even in the infinite layer limit, atleast information about connectivity in terms of reaches will be retained.
> >
> > Additionally, we suspect that if some form of oversmoothing occurs, its “convergence speed” will be slower: Investigating  $e^{-\Delta \cdot t}$ from the spectral-response perspective introduced in our paper, we see that additional non-zero terms appear in the spectral decomposition of $e^{-\Delta \cdot t}$. These act counter to an overly-fast loss of information.
> >
> > As a fully rigorous exploration of this topic would constitute a research paper in its own right, we leave further investigation of these ideas (beyond an additional empirical ablation studywhich we are performing and will include) for future research.
> >
> >
> > References
> >
> > [1]: Chaoqi Yang, Ruijie Wang, Shuochao Yao, Shengzhong Liu, Tarek Abdelzaher, Revisiting Over-smoothing in Deep GCNs, https://arxiv.org/abs/2003.13663
> >
> > [2]: J. J. P. Veerman and Robert Lyons, A Primer on Laplacian Dynamics in Directed Graphs, Nonlinear Phenomena in Complex Systems,  2020, https://api.semanticscholar.org/CorpusID:211066395

---

> ### Comment · Reviewer_T8Vv · 2023-11-20
> **Response to the authors**
>
> Thank you for addressing my questions. I believe it's a good paper and vote for acceptance.

---

### Official Review · Reviewer_RxaJ · 2023-10-30

**Soundness:** 3 good
**Presentation:** 3 good
**Contribution:** 3 good
**Rating:** 6
**Confidence:** 3

**Summary:**

This paper introduces a novel approach to extending spectral graph neural networks (GNNs) from undirected graphs to directed graphs.
Instead of leveraging the traditional Graph Fourier Transform, this paper employs the holomorphic function to avoid the self-adjoint requirement. This paper further discusses the significance of this extension, highlighting its frequency-response interpretation and the basis used to express filters. Finally, the proposed method, namely FaberNet, is validated through experiments on diverse datasets.

**Strengths:**

1. This paper provides a new direction for designing spectral GNNs in directed graphs, i.e., holomorphic functions, which can preserve the algebraic relations to construct the polynomial filters.

2. The authors give sufficient justifications to verify the advantages of holomorphic functions, convincing me about their effectiveness.

3. The proposed FaberNet consistently outperforms other directed GNNs in both heterophilic node classification and regression tasks.

**Weaknesses:**

I'm concerned about the datasets used in the node classification task. Results in Table 1 show that the performance of MagNet is far way from FaberNet. I guess that this is because MagNet operates as a low-pass filter and therefore cannot perform well in the heterophilic datasets.

Since extending spectral GNNs to direct graphs mainly relies on the modification of eigenvectors, I think the choice of filters should not be the major reason to affect the model performance.

It would be better if the authors could provide more experimental results in the homophilic graphs to validate the effectiveness of the proposed methods.

**Questions:**

Replacing $\lambda$ with $T$ yields equation 2. I wonder what $d$ indicates in $T - z \cdot I d$? This symbol seems to conflict with the differential symbol.

---

> ### Author Response · Authors · 2023-11-18
> **Provdided an extended Discussion of the Interplay between Filter-Choice, Operator Selection and Dataset-specific-Performance + Are running additional Experiments on homophilic Graphs**
>
> We would like to sincerely thank the reviewer for the careful consideration of our paper! We were especially happy to read that we
> > give sufficient justifications to verify the advantages of holomorphic functions, convincing [the reviewer] about their effectiveness”
>
> and it was appreciated by the reviewer that our approach
> > consistently outperforms other directed GNNs in both heterophilic node classification and regression tasks”.
>
> Let us address the raised points individually below:
>
>
>
> * "Results in Table 1 show that the performance of MagNet is far way from FaberNet. I guess that this is because MagNet operates as a low-pass filter and therefore cannot perform well in the heterophilic datasets."
>
> We personally do not believe that the observed performance gap necessarily arises because MagNet operates as a low pass filter:
>
> In standard graph convolutional architectures, the effect of (low-pass) Laplacian smoothing indeed makes node representations more and more similar as a signal is propagated through the layers (c.f. e.g. [1]). This is then especially bad in the heterophilic setting, where node-labels of a given node and those of its neighbourhood differ.
>
> Mathematically, Laplacian smoothing can be understood as mapping an original feature matrix $X$ to its smoothed version $X \mapsto e^{-\Delta \cdot t}\cdot X$. Here $\Delta$ denotes a suitable graph Laplacian and the continuous time $t$ is replaced by the layer-depth $\ell$ when discussing the low-pass behaviour of GNNs. Smoothing now occurs, bevause the function $\lambda \mapsto e^{-\lambda\cdot t}$ suppresses all non-zero eigenvalues $\lambda$.
>
> For large times ($t \gg 1$) or respectively deep architectures, the only information that survives is that corresponding to the $\lambda = 0$ eigenvalue. Since the corresponding eigenvector is (up to normalisation) simply given as $v_0^T = (1,…,1)^T$, this means that only a graph-average  $v_0^T\cdot  X$ of all the available information $X$ survives.
>
> The story is however different when the magnetic Laplacian $\Delta_q$ as in the MagNet Paper is considered: Generically, this Laplacian does not have $0$ as an eigenvalue if $q \neq 0$.
> ((To be exactly precise, $0$ is an element of the spectrum $\sigma(\Delta_q)$ if and only if the magnetic potential that gives rise to $\Delta_q$ is gauge-equivalent to the trivial potential; c.f. e.g. [2].))
> Since $0$ is (generically) not an eigenvalue of $\Delta_q$, we have that $e^{-\Delta_q\cdot t}$ does not (generically) converge to a graph average, when  $t$ (respectively the layer-depth) is increased.
> This makes us question, whether the MagNet architecture really acts as a low-pass filter.
>
> Our interpretation of the performance gap between MagNet and FaberNet is a different one:
>
> The operator on which MagNet is based, is the magnetic Laplacian $\Delta_q$. Up to magnetic modifications and normalization, this operator is of the form $D – A$, with $D$ the degree matrix and $A$ the adjacency matrix. Applying this operator to the feature matrix $X$, thus essentially compares the local feature of a given node (retained via $D$) with the features of all surrounding nodes (aggregated via $A$).
>
> As was remarked in "Improving Graph Neural Networks with Simple Architecture Design” (i.e. “Maurya et al., 2021” in our Bibliography) the following holds:
> > Under homophily, nodes are assumed to have neighbors with similar features and labels. Thus, the cumulative aggregation of node’s self-features with that of neighbors reinforce the signal corresponding to the label and help to improve accuracy of the predictions. While in the case of heterophily, nodes are assumed to have dissimilar features and labels. In this case, the cumulative aggregation will reduce the signal and add more noise causing neural network to learn poorly and causing drop in performance. Thus it is essential to have node’s self-features separate from the neighbor’s features.
>
> To avoid comparing the features of surrounding nodes with the feature vector of a given node our architecture is based on (a modified version of) only the adjacency matrix $A$ (instead of a Laplacian $\sim D - A $).
> The adjacency matrix has only the entry $0$ on the diagonal. Thus, when updating the feature vector of a given node, the previous feature vector of this node is discarded, and the new feature vector is solely made up of information about surrounding nodes (i.e. information about the neighbourhood(structure) of the original node). This avoids  mixing and comparing  self-features with neighbourhood features. This is beneficial in the heterophilic setting, as (Maurya et al., 2021) pointed out theoretically, and we also observed empirically.

---

> > ### Author Response · Authors · 2023-11-18
> > **The influence of Filters on Model Performance and conducting Experiments on heterophilic vs. homophilic Graphs**
> >
> > * "Since extending spectral GNNs to direct graphs mainly relies on the modification of eigenvectors, I think the choice of filters should not be the major reason to affect the model performance."
> >
> > In the spirit of academic discussion, we would like to respectfully disagree with these two points and are hoping we will be able to convince the reviewer of our views on the matter:
> >
> > It is certainly valid that in the undirected setting, and for a fixed characteristic operator, the effect of applying a spectral filter may be described purely in terms of scalar modifications of eigenvectors of said characteristic operator.
> >
> > As we point out in Section 2 and discuss in additional detail in Appendix B, this is no longer the case in the directed setting: There simply does no longer exist a complete Basis of eigenvectors associated to a given characteristic operator (such ass an adjacency matrix or a graph-Laplacian). Thus the effect of a spectral filter may not be completely described purely in terms of effects on eigenvectors. In fact, there generically only needs to exist a single true eigenvector, so that such a description would be severely incomplete.
> >
> > Instead -- as we argue in the paper – one needs to transcend precisely this reliance on eigenvectors (or equivalently the graph Fourier transform), when extending spectral convolutions to the directed setting.
> > The appropriate replacement is to instead focus on learnable functions $g$ applied to characteristic operators $T$ as $g(T)$.
> > It is then precisely the interplay of the characteristic operator $T$ with the learnable filter functions $g$ that shapes the inductive bias within the model and determines its performance on any given task:
> >
> > While addressing an earlier concern raised by the reviewer, we have already discussed how the choice of characteristic operator $T$ determines the performance of the model on heterophilic vs. homophilic graphs: On heterophilic graphs, it is detrimental to utilize characteristic operators $T$ that intrinsically and directly compare individual node features with that of  surrounding nodes, such as e.g. graph Laplacian do. Here it is better to base the model on some form of the adjacency matrices.
> >
> > As an additional example of the importance of the choice of  filter functions and characteristic operators, consider the Dir-ResolvNet architecture proposed in addition to FaberNet within our paper. This architecture produces graph-level feature vectors that are insensitive to the fine-print articulation of graphs, which allows it to remain stable under topological perturbations. This is achieved precisely because filters are based on the un-normalized graph Laplacian and filter-functions go to zero if the (modulus of) the input argument becomes large: Un-normalized graph Laplacians encode the fine-print information of graphs into large eigenvalues and precisely this information is then suppressed by our choice of filter functions.
> > Thus the choice of filters has a profound influence on the performance of a spectral model on any given task.
> >
> > * It would be better if the authors could provide more experimental results in the homophilic graphs to validate the effectiveness of the proposed methods.
> >
> > In (Rossi et. al., 2023) it was recently established that considering directionality within graphs is especially beneficial in the heterophilic regime. Hence we focused on such graphs when designing our experiments and -- as described in the response to the raised points above -- our method is specifically designed with the heterophilic setting in mind. As such we found the choice of datasets to be appropriate.
> >
> > Nevertheless, we agree with the reviewer that it would be interesting to also consider the method’s performance in the homophilic setting, even if it was not designed for this setting.
> >
> > The corresponding experiments are still ongoing and the full results will be included in a camera-ready version of this paper. Nevertheless, we here already provide preliminary resuls on the two standard datasets Cora and Citeseer for FaberNet and both a representative direced- as well as undirected baseline:
> >
> > | [\%]   |    DiGCN |GCN   | FaberNet  |
> > |---|---|---|---|
> > |  Cora| 80.28 $\pm$ 0.51  |71.01 $\pm$ 0.37   | 75.64 $\pm$ 0.69  |
> > |   Citeseer|  66.02 $\pm$ 0.52 | 63.12 $\pm$ 0.35 |  65.51 $\pm$ 0.37  |
> >
> > FaberNet (being designed for the heterophilic setting) does not ouperform DiGCN on these  homophilic graphs, while it still performs significantly better than GCN. As a Caveat, it should be noted that (making use of the DiGCN-codebase) optimized hyperparamters were used for both GCN and DiGCN. For FaberNet a simple fixed $64 \times 64$ architecture was used and a full tuning of hyperparameters has not been completed yet. We will include an updated table with added baselines and additional datasets in a camera ready version of our paper.

---

> > > ### Author Response · Authors · 2023-11-18
> > > **Addressing the Question on Notation and References**
> > >
> > > * "Replacing $\lambda$ with $T$ yields equation 2. I wonder what $d$ indicates in $T - z \cdot Id$? This symbol seems to conflict with the differential symbol.
> > >
> > > In our response, we are assuming that the reviewer is referring to the term $dz$ in equations (1) and (2). We hope that we understood this correctly; if not we would be very glad to provide further clarifications.
> > >
> > > The “dz” term represents the line-integration-measure in the complex plane. To clarify this point raised by the reviewer, we have added the sentence “Here "$dz$" denotes the complex line integration measure in $\mathbb{C}$.”  in the discussion of equations (1) and (2) in our revised manuscript.
> > >
> > > References:
> > >
> > > [1]: Chaoqi Yang, Ruijie Wang, Shuochao Yao, Shengzhong Liu, Tarek Abdelzaher, Revisiting Over-smoothing in Deep GCNs, https://arxiv.org/abs/2003.13663
> > >
> > > [2]: John Stewart Fabila, The discrete magnetic Laplacian: geometric and spectral preorders with applications (PhD Thesis), https://www.icmat.es/Thesis/2020/Tesis_Jhon_Stewart_Fabila_Carrasco.pdf

---

### Official Review · Reviewer_tzrV · 2023-11-01

**Soundness:** 3 good
**Presentation:** 2 fair
**Contribution:** 4 excellent
**Rating:** 6
**Confidence:** 3

**Summary:**

This paper studied the problem of designing graph filters for directed graphs. Instead of following the same approach for undirected cases (first define the spectrum based on Graph Fourier Transform, then define the filters based on spectral responses), the authors proposed to analyze the relationship between spectral response $g(\lambda)$ and graph filter $g(T)$ directly when function $g$ comes from some constrained set. By observing an interesting connection between this problem and results from functional calculus, this paper presented a principal design for directed graph filters and two feasible approximations: faber polynomials and resolvent. Furthermore, the stabilities of these graph filters are established, and the simulations show that the new method outperforms other directed GNN baselines over several benchmark datasets.

**Strengths:**

I believe the most important contribution of this paper is that the authors point out the fact that we do not really need Graph Fourier Transform to define graph filters, which has always been an obstacle in graph learning and signal processing community due to the non-symmetric nature of directed Laplacian. The literature review and observations presented in this paper can be viewed as a general guidance to design convolutional kernels for directed graphs, and lay the theoretical foundations for other future works on this subject. Following the general design, this paper also introduced two computationally feasible filter banks to implement the directed graph kernels in practice, which achieves good results on standard benchmark datasets.

**Weaknesses:**

Although the technical insights shown in this paper are great and very interesting, the presentation of this paper, especially the main text, can be improved. I understand that the authors want to convey as much information as possible within the main text, but the main goal of this paper, which is the implementation of the practical directed convolutional kernels (layers), should still be explained clearly in detail. For example, after reading the entire paper, I still do not know if we need to perform the eigendecomposition of the graph operator or not. In other words, even if we set the function to be faber polynomials $\Psi_k(\lambda)=\lambda^k/2^k$, what will $\Psi_k(T)$ in this case, where $T$ is the operator? Do we still have to consult Eq (3) to compute $\Psi_k(T)$, which can be computationally expensive? This type of question should be clearly explained in the main text to help readers better understand the big picture. A pseudocode of the algorithm/training procedure can help solve this issue.

Besides the computation of graph kernels, some of the heuristic choices adopted in Holonets lack theoretical insights as well. For example, why do we need to use forward and backward filters separately? Why the operator is chosen to be $T=(D^{in})^{-1/4}W(D^{out})^{-1/4}$? Why use absolute value as the nonlinear activation, which is unusual in GNN literature? All of these choices seem to just come out of trial and error in simulations. It would be good if the authors could provide some further explanations for these choices.

One minor concern that I have is about the simulation of directed graph regression problems. I am not very convinced why we should treat this as a directed graph problem, and how it helps when we consider the edges to be directed and the nodes to be weighted. Maybe it has some physical/chemical meanings, but since I am not an expert in physics, I will leave this to other reviewers to decide.

Two other small comments: I would not describe the difference between the performance of FaberNet and that of Dir-GNN to be significant, as most of the differences are within one standard deviation. Also from the implementation side, Dir-GNN is much simpler than FaberNet; on page 5, "Faber polynomials provide near near mini-max polynomial approximation" there are two "near".

**Questions:**

Besides the ones listed above, I have the following questions as well:

1. What is the choice of $y$ for Dir-ResolvNet?

2. Why is the normalization for $T$ $-1/4$ not $-1/2$? If in undirected case, I believe the practice is to use $D^{-1/2}$. Why is there a discrepancy? Also, I am not sure how this is related to the motivation stated in the paper "We thus use as characteristic operator a matrix that avoids direct comparison of feature vectors of a node with those of immediate neighbours".

3. Where do the imaginary weights in FaberNet come from? Is it because the eigenvalues can be complex?

---

> ### Author Response · Authors · 2023-11-18
> **Followed the received advice dilligently, answered raised questions and added more detail to discussions of architectures/design choices**
>
> We would like to sincerely thank the reviewer for the careful review of our paper! We were especially happy to read that
> > the technical insights shown in this paper are great and very interesting
>
>  and that
> >[our paper] lay[s] the theoretical foundations for other future works on [convolutional kernels for directed graphs].
>
> Let us address raised points individually:
>
> * "Although the technical insights shown in this paper are great and very interesting,  [...] the implementation of the practical directed convolutional kernels (layers), should still be explained clearly in detail."
>
> We agree that our paper is written to serve a dual purpose: To --on the one hand-- develop the theory of spectral convolutions on directed graphs in detail so that it may be  taken up by the community for further research and on the other hand to establish our own specific architectures conforming to the developed framework.
>
> Following this advice by the reviewer, we have now put additional emphasis on introducing and describing our specific architectures in more detail, as we explain below.
>
>
> * "For example, after reading the entire paper, I still do not know if we need to perform the eigendecomposition of the graph operator or not."
>
> We thank the reviewer for this important and indeed central question.  The answer is that an explicit eigendecomposition (or in fact a Jordan Chevalley decomposition in the directed setting) NEVER needs to be computed in our approach.
>
> Following this question by the reviewer, we have now modified the final paragraph of the corresponding Section 3.2 in our updated manuscript. It now reads:
> > “For us, the spectral response (4) provides guidance when considering scale-insensitive convolutional filters on directed graphs in Sections 3.3 and 4 below. The spectral response (4) is however never used to _implement_ filters: As discussed above, this is achieved much more economically via (3).”
>
> * "[...] if we set the function to be faber polynomials $\Psi_k(\lambda) = \lambda^k/2^k$, what will $\Psi_k(T)$ in this case, where $T$ is the operator? "
>
> If we have $\Psi_k(\lambda) = \lambda^k/2^k$ then we exactly have
>
> $$ \Psi_k(T) = \frac{1}{2^k}T^k.$$
>
> We had initially discussed this effect of applying the holomorphic functional calculus to monomials and polynomials at the end of Section 3.1. Following this valid point raised by the reviewer, we have -- for additional clarity -- promoted said discussion at the end of Section 3.1 into a Theorem (namely Theorem 3.1 in our updated manuscript) into its own right for heightened visibility.
>
> Additionally, we now also explicitly provide the above result of applying the Faber polynomial $\Psi_k(\lambda)$ to the operator $T$ already in Section 3.3.1 (i.e. precisely when Faber Polynomials are first introduced).
>
>
>
> * "Do we still have to consult Eq (3) to compute $\Psi_k(T)$, which can be computationally expensive? "
>
> The answer to this question is a resounding NO:
>
> Equation (3) never needs to be used to implement a filter in practice. The purpose of this equation is instead to serve as a theoretical tool to spectrally interpret the action of given filters. Additionally it provides guidance on the choice of suitable filters adapted to a given task.
>
> To avoid a costly explicit Jordan-Chevalley decomposition, our model instead uses parametrized spectral convolutional filters, which were introduced and discussed in the first paragraph of Section 3.4.
>
> This is indeed an important point brought up by the reviewer, and – as already mentioned above – we now explicitly state the following in our revised manuscript
> [Note that equation (4) in the revised document refers to equation (3) in the original submission]:
> >“For us, the spectral response (4) provides guidance when considering scale-insensitive convolutional filters on directed graphs in Sections 3.3 and 4 below. The spectral response (4) is however never used to _implement_ filters: As discussed above, this is achieved much more economically via (3).”.
>
> * "This type of question [as above] should be clearly explained in the main text to help readers better understand the big picture."
>
> We hope that the provided modifications in our updated manuscript have clarified these points. Should this not yet be the case, we will of course be more that happy to act on any additional feedback.
>
> * "pseudocode of the algorithm/training procedure can help solve this issue."
>
> Apart from the aforementioned added and modified discussions, we have also included a pseudocode of the forward of our model  in our revised manuscript (c.f. Appendix J) which we reference in the main text of our revised manuscript.

---

> > ### Author Response · Authors · 2023-11-18
> > **Discussing Design Choices I**
> >
> > * "Besides the computation of graph kernels, some of the heuristic choices adopted in Holonets lack theoretical insights as well. "
> >
> > We will be glad to shed additional light on these design choices below.
> >
> >  To clarify these points also in out paper, we have introduced an additional paragraph “Design Choices” in “Appendix I: Additional Details on Experiments”, where we include  of the discussions below.
> >
> > * "For example, why do we need to use forward and backward filters separately?"
> >
> > In principle, the need to consider forward and backward filters arises from the nature of directed graphs: If – say-- only forward filters would be implemented, information would only be allowed to flow _along_ the directed edges of a given graphs.
> >
> >  In a citation network (with the characteristic operators $T$ chosen e.g. as the adjacency matrix or a graph Laplacian) for example, this would only endow a given node representing a specific paper with information about papers that the given paper cites. The (equally important) information about papers that cite the given one never reaches the node of our fixed paper.
> > This can be remedied by not only considering filters based on $T$, but also its transpose $T^*$: One can consider $T^*$ as describing the original graph, with all edge directions reversed, so that $T^*$ facilitates information flow in exactly the opposite direction when compared with $T$.
> >
> > In principle, one could then be tempted try to implement filters as functions $f(x,y)$ in two complex variables ($x$ and $y$) and make the replacements $z \mapsto T$ and $y \mapsto T^*$, thus implementing a joint filter $f(T, T^*)$ for both directions simultaneously.
> >
> > However, such a procedure is unfortunately mathematically ill-defined: Consider for example the function $f(z,y) = zy = yz$. It is then not clear if we should set $f(T, T^*) = T \cdot T^*$ or $f(T, T^*) = T^* \cdot T$.
> >
> >  This would not be a problem if $T^* \cdot T = T \cdot T^*$ and one may indeed prove that in this setting, one might consistently define the matrix $f(T, T^*)$; the relevant tool here is the “Functional calculus for commuting operators”.
> >
> > In general however, we have $T^* \cdot T \neq T \cdot T^*$ if $T$ describes a directed graph. Hence we can not construct such a joint spectral filter $f(T, T^*) $ and instead consider learnable spectral filters of the form $f(T) + g(T^*)$. In principle, one may of course also consider other algebraic combination of the two directions of information flow (such as e.g. $f(T) \cdot g(T^*)$ and $g(T^*)\cdot f(T)$). We however leave this for future work.
> >
> > *  Why the operator is chosen to be $T = (D^{\text{in}})^{-\frac14}W(D^{\text{out}})^{-\frac14}$? [...]
> >  Also, I am not sure how this is related to the motivation stated in the paper "We thus use as characteristic operator a matrix that avoids direct comparison of feature vectors of a node with those of immediate neighbours".
> >
> > As we understand it, there are two aspects to these questions:
> > 1) Why is the characteristic operator chosen as some form of (normalized) adjacency matrix (as opposed to –say-- a Laplacian or a renormalized version of the adjacency matrix (i.e. with self-loops) as e.g. in the GCN paper by Kipf & Welling)?
> >
> > Our aim is to build a network that performs well for node classification on heterophilic graphs.
> > To this end, the following has been noted in "Improving Graph Neural Networks with Simple Architecture Design” (i.e. “Maurya et al., 2021” in our Bibliography):
> > >Under homophily, nodes are assumed to have neighbors with similar features and labels. Thus, the cumulative aggregation of node’s self-features with that of neighbors reinforce the signal corresponding to the label and help to improve accuracy of the predictions. While in the case of heterophily, nodes are assumed to have dissimilar features and labels. In this case, the cumulative aggregation will reduce the signal and add more noise causing neural network to learn poorly and causing drop in performance. Thus it is essential to have node’s self-features separate from the neighbor’s features.
> >
> > To avoid comparing the features of surrounding nodes with the feature vector of a given node when applying the characteristic operator $T$ (or powers $T^k$ of $T$) to the feature matrix $X$ as $X \mapsto T\cdot X$, we choose $T$ to have only the entry $0$ on the diagonal. Thus, when updating the feature vector of a given node, the previous feature vector of this node is discarded, and the new feature vector is solely made up of information about surrounding nodes (i.e. information about the neighbourhood(structure) of the original node). This avoids a mixing of self-features and neighbourhood features, which is desireable as (Maurya et al., 2021) pointed out theoretically, and we also observed empirically in our experiments. Hence we choose $T$ as (some form of) the adjacency matrix, as it has a zero-diagonal.

---

> ### Author Response · Authors · 2023-11-18
> **Discussing Design Choices II and Providing additional Details on Experiments**
>
> 2) Why is the normalization chosen as $D^{-frac14}$ as opposed to $D^{-frac12}$ ?
>
> It is indeed true that a symmetric normalization by $D^{-frac12}$ is traditional in the undirected setting. Historically, this can be traced back to the utilization of the symmetrically normalized Laplacian as providing the Fourier-atoms (i.e. the Laplacian eigenvectors) of the graph Fourier Transform that underlies e.g. the original undirected spectral models such as (Bruna et al., 2014), (Defferrard et al., 2016) (Kipf & Welling, 2017) [using the nomenclature of the Bibliography of our submission].
>
> A main point of the present paper however, is transcending this graph Fourier transform, as the traditional approach is limited to undirected graphs. Our approach instead does not need to be based on any underlying Laplacian on the graph: Indeed, the holomorphic functional calculus may be applied to arbitrary operators.
>
> As such, a traditional holding-on-to a normalization stemming from a traditional use of a Laplacian-based Graph-Fourier transforms is no longer necessary and we want to point this out to the community. As we observed experimentally, such a traditional normalization is also (slightly) disadvantageous, as far as performance is concerned.
>
>
> * "Why use absolute value as the nonlinear activation, which is unusual in GNN literature?"
>
> It is indeed true that a large percentage of the machine learning literature in general uses some form of ReLU activations. In the graph setting, an exception is constituted by graph-scattering-networks (c.f. e.g. [1,2]): Corresponding scattering architectures are based on the  absolute-value-nonlinearity.
>
> For us, the choice to use either ReLu or the absolute value arises as our networks are potentially complex. In the complex setting there is no clear consensus which activation function performs best (c.f. e.g. [3]). Hence we consider two possible choices in our architecture, whenever we enable complex parameters (c.f.also Table 7 in Appendix I)
>
>
> * "All of these choices seem to just come out of trial and error in simulations. It would be good if the authors could provide some further explanations for these choices."
>
> We hope that we were able to clarify these points. As mentioned above, we have added corresponding discussions of the points raised by the reviewer in our revised manuscript.
>
>
> * "One minor concern that I have is about the simulation of directed graph regression problems. I am not very convinced why we should treat this as a directed graph problem, and how it helps when we consider the edges to be directed and the nodes to be weighted. Maybe it has some physical/chemical meanings, but since I am not an expert in physics, I will leave this to other reviewers to decide."
>
> Our view of this matter is as follows:
>
> Our aim in this present paper is to establish within the graph learning community that the use of spectral methods on directed graphs is not limited to achieving  state of the art performance on node-classification: HoloNets can e.g. also be used at the graph level, to construct networks that are transferable between directed graphs describing the same underlying object at different resolution scales.
>
> While transferability is certainly an established research topic for graph neural networks, the approach via of such multiscale-consistency has only recently started to be investigated. As such standard benchmarks have not yet been established; especially not in the directed setting. We thus modify a standard dataset (i.e. QM7) that was recently used to test for multiscale-consistency in the undirected setting (c.f. Koke et al. (2023)) in order to test for such consistency in the directed setting.
>
> Whether a molecule is represented as a directed or undirected graph is irrelevant from the perspective of retained physical information: Both descriptions may be translated into each other.
> In either case, the node-weights arise naturally as a way to represent additive (under combining nodes) node-wise information, such as charges (like in the case at hand), particle numbers (e.g. Neutron- or Proton numbers) or masses corresponding to individual nodes/atoms.
>
> * "I would not describe the difference between the performance of FaberNet and that of Dir-GNN to be significant, as most of the differences are within one standard deviation. Also from the implementation side, Dir-GNN is much simpler than FaberNet; "
>
> That is a valid point. We have amended the corresponding statement about the comparison between Dir-GNN and Faber-Net in Section 5.1.
>
> * "on page 5, "Faber polynomials provide near near mini-max polynomial approximation" there are two "near"."
>
> We thank the reviewer for spotting this typo, which we have now correced.

---

> ### Author Response · Authors · 2023-11-18
> **Discussing Final Questions and References**
>
> * "What is the choice of $y$  for Dir-ResolvNet?
>
> We have chosen $y = -1$ for simplicity. We had already detailed this in “Appendix I: Additional Details on Experiments”, but have now also included this in the main text of our paper in our revised manuscript.
>
> * "Where do the imaginary weights in FaberNet come from? Is it because the eigenvalues can be complex?"
>
> Yes, that can indeed be thought of as the underlying reason.
>
> In more detail: If the underlying graph is directed, the associated characteristic operators $T$ are generically not self-adjoint. Hence eigenvalues of such operators are generically complex. If one intends to apply a function $g$ to such a matrix $T$, this necessitates $g$ to be defined at least in a neighbourhood of each eigenvalue of $T$, as can e.g. be seen from the spectral response discussed in the paper. Thus $g(z)$ needs to be defined for complex $z$. If one represents such a function via simpler functions (e.g. a polynomial $g(z)$ represented via a sum of monomials $z^k$ as $g(z) = \sum_k a_k z^k$ ), the corresponding coefficients $a_k$ are generically complex. These coefficients precisely constitute the learnable parameters in our method.
>
> References:
>
> [1]: Michael Perlmutter, Feng Gao, Guy Wolf, and Matthew Hirn. Understanding graph neural networks with asymmetric geometric scattering transforms, 2019. URL https://arxiv.org/abs/1911.06253.
>
> [2]: Christian Koke and Gitta Kutyniok. Graph scattering beyond wavelet shackles. In Advances in Neural Information Processing Systems 35: Annual Conference on Neural Information Processing Systems 2022, NeurIPS 2020, November 28 - December 9, 2022, New Orleans. OpenReview.net, 2023. URL https://openreview.net/forum?id=ptUZl8xDMMN.
>
> [3]: ChiYan Lee, Hideyuki Hasegawa, and Shangce Gao. Complex-valued neural networks: A comprehensive survey. IEEE/CAA Journal of Automatica Sinica, 9(8):1406–1426, 2022. doi:10.1109/JAS.2022.105743.

---

> > ### Comment · Reviewer_tzrV · 2023-11-22
> >
> > I really appreciate the authors' effort to prepare this detailed response. It took me a bit long to read through all the content. I would suggest next time the authors can use a different color in the revision to mark the modified text. I have no further questions for now.

---

### Meta-Review · Area_Chair_1Drb · 2023-12-08

**Metareview:**

In this submission, the authors propose a new member of spectral GNNs called HoloNet, extending spectral GNN to modeling directed graphs. The authors conducted a sufficient analysis of the proposed model, demonstrating its rationality and providing useful insights. Experimental results demonstrate the effectiveness of the proposed model.

Strengths: (a) The proposed method is reasonable, whose rationality is supported by theory, and the authors provided detailed analysis. (b) Experimental results show the superiority of the proposed method clearly.

Weaknesses: The presentation of the submission should be enhanced.

**Justification For Why Not Higher Score:**

The authors proposed a solid method to extend spectral GNN to modeling directed graphs. The authors' claims are supported by both theoretical analysis and experimental results. However, the topic itself (i.e., modeling directed graphs via spectral methods) seems to be a sub-topic of the GNN study, which may only attract a part of researchers in the community. Additionally, the presentation of this work should be enhanced, making it more friendly to the researchers without sufficient background.

**Justification For Why Not Lower Score:**

All reviewers provide positive feedback.

---

### Decision · Program_Chairs · 2024-01-16

Accept (poster)